# Hepatic HuR modulates lipid homeostasis in response to high-fat diet

Zhuojun Zhang [1,13], Chen Zong[1,13], Mingyang Jiang[2], Han Hu [1], Xiaolei Cheng[1], Juhua Ni[1], Xia Yi[1], Bin Jiang [1], Feng Tian[3], Ming-Wen Chang[4], Wen Su[5], Lijun Zhu [6], Jinfan Li[7], Xueping Xiang [7], Congxiu Miao [8], Myriam Gorospe[4], Rafael de Cabo [9], Yali Dou [10], Zhenyu Ju [11], Jichun Yang [12], Changtao Jiang [12], Zhongzhou Yang [2✉] & Wengong Wang [1✉]

Lipid transport and ATP synthesis are critical for the progression of non-alcoholic fatty liver disease (NAFLD), but the underlying mechanisms are largely unknown. Here, we report that the RNA-binding protein HuR (ELAVL1) forms complexes with NAFLD-relevant transcripts. It associates with intron 24 of *Apob* pre-mRNA, with the 3′UTR of *Uqcrb*, and with the 5′UTR of *Ndufb6* mRNA, thereby regulating the splicing of *Apob* mRNA and the translation of UQCRB and NDUFB6. Hepatocyte-specific HuR knockout reduces the expression of APOB, UQCRB, and NDUFB6 in mice, reducing liver lipid transport and ATP synthesis, and aggravating high-fat diet (HFD)-induced NAFLD. Adenovirus-mediated re-expression of HuR in hepatocytes rescues the effect of HuR knockout in HFD-induced NAFLD. Our findings highlight a critical role of HuR in regulating lipid transport and ATP synthesis.

[1] Department of Biochemistry and Molecular Biology, Beijing Key Laboratory of Protein Posttranslational Modifications and Cell Function, School of Basic Medical Sciences, Peking University Health Science Center, 38 Xueyuan Road, 100191 Beijing, China. [2] State Key Laboratory of Pharmaceutical Biotechnology and MOE Key Laboratory of Model Animal for Disease Study, Model Animal Research Center, Nanjing Biomedical Research Institute, Nanjing University, 210061 Nanjing, China. [3] Department of Laboratory Animal Science, Peking University Health Science Center, 38 Xueyuan Road, 100191 Beijing, China. [4] Laboratory of Genetics and Genomics, National Institute on Aging, National Institutes of Health, 251 Bayview Blvd., Baltimore, MD 21224, USA. [5] Department of Pathology, Shenzhen University Health Science Center, 519060 Shenzhen, China. [6] Zhejiang Provincial Key laboratory of Pancreatic Diseases, the First Affiliated Hospital, School of Medicine, Zhejiang University, 310006 Hangzhou, China. [7] Department of Pathology, the Second Affiliated Hospital, School of Medicine, Zhejiang University, 310009 Hangzhou, China. [8] Institute of Reproduction and Genetics, Changzhi Medical College, 046000 Changzhi, China. [9] Translational Gerontology Branch, National Institute on Aging, National Institutes of Health, Baltimore, MD 21224, USA. [10] Department of Pathology and Biological Chemistry, University of Michigan, 1301 Catherine Street, Ann Arbor, MI 48105, USA. [11] Key Laboratory of Regenerative Medicine of Ministry of Education, Institute of Aging and Regenerative Medicine, Jinan University, 510632 Guangzhou, China. [12] Department of Physiology and Pathophysiology, School of Basic Medical Sciences, Peking University Health Science Center, 38 Xueyuan Road, 100191 Beijing, P.R. China. [13] These authors contributed equally: Zhuojun Zhang, Chen Zong. ✉email: zhongzhouyang@nju.edu.cn; wwg@bjmu.edu.cn

Non-alcoholic fatty liver disease (NAFLD) is associated with a variety of disease conditions including obesity, insulin resistance, diabetes, hypertension, hyperlipidemia, and metabolic syndrome[1]. Characterized by the accumulation of lipids in liver, NAFLD results from the imbalance between lipid acquisition and disposal[2–4]. Excessive intake of dietary fat, abnormal lipid synthesis, and liver oxidation increase the levels of liver lipids[2–4], which can then be eliminated by two major paths: β-oxidation of fatty acids into acetyl-coenzyme A and to enter the Krebs cycle and generate ATP, and export of liver lipids to other tissues. Failure to dispose excess liver lipids by one of these mechanisms leads to NAFLD[5,6].

Several different factors are involved in regulating hepatic lipid transport and ATP synthesis. PPARα (peroxisome proliferator-activated receptor, PPARα) is a key transcriptional regulator of fatty acid oxidation in mitochondria[7,8]. Oxidation of fatty acids occurs mainly in the mitochondria, generating ATP through oxidative phosphorylation through the action of proteins encoded by both mitochondrial and nuclear DNA[6]. Alterations in these factors leading to aberrant ATP production in NAFLD have been documented in patients[9,10], as well as in mice and rats with NAFLD[11,12]. Impaired ATP synthesis in NAFLD arises from the reduced activity or levels of factors in complexes I–V[9]; for complex II, since there are no factors encoded by mitochondria DNA, deficiencies in the activity of this complex reflect aberrant expression of factors encoded from nuclear DNA. On the other hand, when liver lipids are in excess, they can be packaged into VLDL particles, transported to serum and distributed to other tissues[5,13].

Apolipoprotein B-100 (APOB-100) and apolipoprotein E (APOE) are critical for the packaging and secretion of VLDL to maintain hepatic lipid homoeostasis[14,15]. APOB abundance is altered in NAFLD, suggesting a role for APOB in this disease. APOB production is regulated at multiple levels[16–19]. Transcription of the *Apob* gene is governed by transcription factors HNF-4, HNF-3β, ARP-1, and C/EBPβ[16,17], while APOB translation is controlled by RNA-binding proteins (RBPs) and microRNAs interacting with *Apob* mRNA[18]. While the splicing of *Apob* pre-mRNA has been targeted therapeutically for lowering blood cholesterol levels[19], the mediators of this regulation are unknown. HuR ['human antigen R', also known as ELAVL1 (embryonic lethal abnormal vision-like 1)], is a ubiquitous member of Hu/ELAV RBP family. HuR regulates the post-transcriptional fate of many coding and noncoding RNAs[20–22], in turn regulating many cell functions (proliferation, survival, apoptosis, senescence, and differentiation) and affecting processes such as cancer and aging. HuR was also reported to promote ATP synthesis in cells by regulating the translation of cytochrome c (CYCS) and coenzyme Q7 (COQ7)[23,24]. A recent study describes HuR as a regulator for ABCA1 translation, influencing macrophage cholesterol metabolism in vivo[25]. However, the role of HuR in lipid metabolism and the underlying mechanisms remains to be studied.

In the present study, a conditional hepatocyte-specific HuR knockout mouse (cKO) is created to evaluate the role of HuR in high-fat diet (HFD)-induced NAFLD. We find evidence that HuR associated with mouse *cytochrome c* (*Cycs*), *Uqcrb*, and *Ndufb6* mRNAs, as well as with *Apob* pre-mRNA, thereby regulating the translation of CYCS, UQCRB, and NDUFB6, as well as the production of *Apob* mRNA. These processes impact upon HFD-induced NAFLD and point to a mechanism whereby HuR controls liver lipid homeostasis.

## Results

### HuR regulates lipid transport and ATP synthesis in NAFLD.
To evaluate the role of HuR in NAFLD, we generated a conditional hepatocyte-specific HuR knockout (cKO) mouse by crossing a *HuR* Flox/Flox mouse (C57BL/6 J background, Jackson Laboratories) with an albumin *Cre* mouse. After confirming the specific knockout of HuR in hepatocytes by real-time qPCR and western blot analyses (Supplementary Fig. 1), we examined liver function in HuR cKO mice and wild-type (WT) littermates fed regular chow. As shown in Fig. 1a, deletion of HuR did not lead to alterations of body weight, liver weight, or the ratio of liver weight relative to body weight. By staining with hematoxylin and eosin (H&E) as well as with Oil Red O, we observed that hepatocytes in cKO mice were comparable morphologically to those of WT littermates (Fig. 1b). Although the levels of liver triglyceride and cholesterol in HuR cKO mice were slightly higher than those observed in WT mice, the difference was not significant (Fig. 1c). Furthermore, hepatocytes in cKO mice displayed slightly reduced levels of ATP, although this reduction was not significant (Fig. 1d). Additional results showed that deletion of HuR reduced the levels of serum APOB ($p < 0.05$) and HDL-C (high-density lipoprotein cholesterol) ($p < 0.01$), but lowered only mildly the serum levels of triglyceride, cholesterol, APOE, LDL-C (low-density lipoprotein cholesterol) (Supplementary Fig. 2a). Overall liver function did not appear to be affected by HuR deletion, as reflected by the unchanged activity of ALT (alanine aminotransferase) and AST (aspartate aminotransferase), and the levels of T-Bil (total bilirubin), ALB (albumin), and TP (total protein) in the serum (Supplementary Fig. 2b).

Lipid uptake, synthesis, oxidation, and transport, as well as ATP synthesis, all are well recognized to influence NAFLD[2–4]. Other studies have reported that HuR regulates the expression of

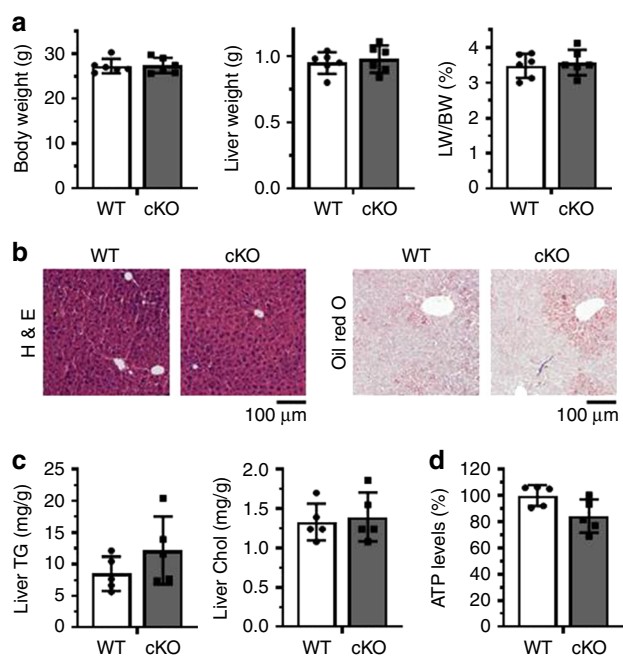

**Fig. 1 Deletion of HuR in the liver does not alter hepatocyte phenotype of liver function. a** Conditional hepatocyte-specific HuR knockout mice (cKO, $n = 6$) and wild-type littermates (WT, $n = 6$) were used for analyzing body weight, liver weight, and the ratio of liver/body weight. **b** Representative images of hematoxylin and eosin (H&E) and Oil Red O staining of liver slices from the mice described in **a** (WT, $n = 5$; cKO, $n = 5$). **c, d** Livers from mice described in **a** were used for analyzing the levels of triglyceride (TG) and cholesterol (Chol) (**c**) (WT, $n = 5$; cKO, $n = 5$), and the levels of ATP (**d**) (WT, $n = 5$; cKO, $n = 5$). Data are the means ± SD. All the error bars are equivalent throughout the figure. Source data are provided as a Source Data file.

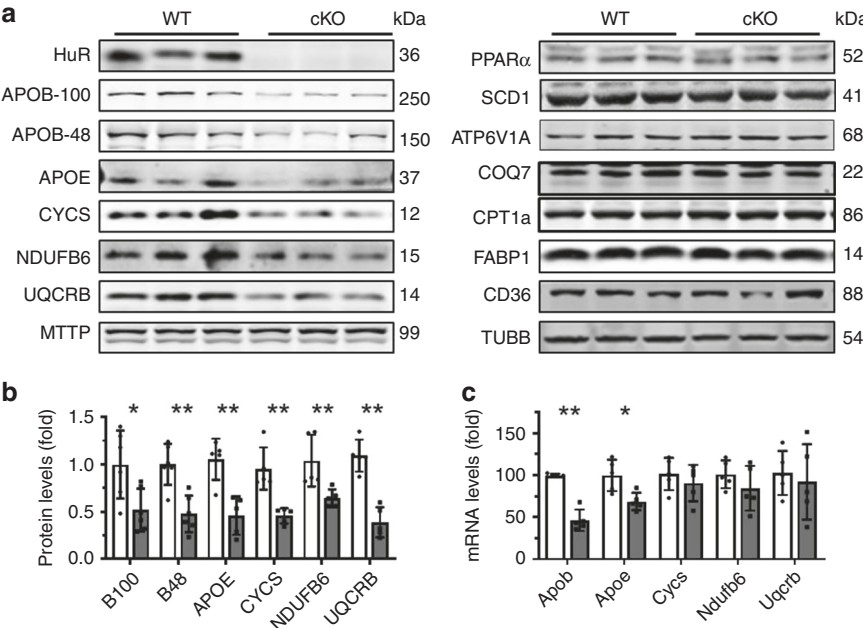

**Fig. 2 Liver HuR ablation reduces the levels of factors that regulate lipid transport and ATP synthesis. a** Protein lysates prepared from liver tissues described in Fig. 1a were subjected to Western blot analysis to assess the levels of proteins HuR, APOB-100, APOB-48, APOE, CYCS, NDUFB6, UQCRB, MTTP, PPARa, ATP6V1A, SCD1, ATP6V1A, COQ7, CPT1a, FABP1,CD36, and β-Tubulin (TUBB). Blots were processed from parallel gels. **b** The density of the signals for APOB-100 (B100), APOB-48 (B48), APOE, CYCS, NUUFB6, and UQCRB [APOB-100 and APOB-48, WT and cKO, $n = 6$; others, WT and cKO, $n = 5$)] in **a** was calculated and plotted as the means ± SD; significance was assessed by using two-tailed Mann–Whitney $U$ test (*$p < 0.05$; **$p < 0.01$). Blank columns, WT; Black columns, cKO. **c** RNA prepared from tissues described in **a** was subjected to RT-qPCR analysis to assess the levels of *Apob*, *Apoe*, *Cycs*, *Ndufb6*, and *Uqcrb* mRNAs (WT and cKO, $n = 5$). Data are the means ± SD (Blank columns, WT; Black columns, cKO); significance was assessed by using two-tailed Mann–Whitney $U$ test (*$p < 0.05$; **$p < 0.01$). All the error bars are equivalent throughout the figure. Source data are provided as a Source Data file.

cytochrome c (CYCS) and COQ7[23,24], even though HuR does not seem to localize in mitochondria (Supplementary Fig. 3a). By bioinformatics analysis (http://starbase.sysu.edu.cn/index.php), HuR was capable of binding the mRNAs that encoded many factors involved in ATP synthesis and transcribed from nuclear DNA (Supplementary Fig. 3b). The levels of proteins NDUFB6 and UQCRB decreased significantly in HuR-deficient livers (Fig. 2a, b). In addition, the levels of APOB-100, APOB-48, and APOE were reduced by HuR ablation (Fig. 2a, b). In agreement with a previous study[23], deletion of HuR reduced the levels of CYCS in liver (Fig. 2a, b), while the levels of proteins MTTP, PPARa, SCD1, CPT1a, ATP6V1A, COQ7, CD36, and FABP1 in liver were unchanged between HuR cKO and WT mice (Fig. 2a). Reverse transcription (RT) followed by real-time quantitative (q) PCR analysis showed that deletion of HuR reduced the levels of *Apob* and *Apoe* mRNAs, but not the levels of *Cycs*, *Ndufb6*, or *Uqcrb* mRNAs (Fig. 2c). Although HuR was capable of regulating the translation of COQ7 in human cells (24), the findings that COQ7 protein levels remained unchanged in HuR cKO mice (Fig. 2a) did not support a regulation of COQ7 by HuR in mouse hepatocytes. Apart from the levels of proteins MTTP, PPARa, SCD1, CPT1a, CD36, and FABP1 (Fig. 2a), the levels of *Srebp1c*, *Elovl6*, *Acaca*, *Fasn*, *Pparγ*, *Ehhadh*, *Acox1*, *Fabp2*, and *Fabp3* mRNAs encoding proteins involving in lipid synthesis, oxidation, or uptake (Supplementary Fig. 4), also remained unchanged in the livers of HuR cKO mice. Taken together, these data suggest that HuR may not be an important regulator for lipid synthesis, oxidation, or uptake, and instead it may regulate lipid transport and ATP synthesis.

To further investigate the role of HuR in NAFLD, HuR cKO mice and their WT littermates were fed with HFD for 4 weeks, whereupon liver function and hepatocyte phenotypes were analyzed. Similar to the mice fed regular chow diet (Fig. 1a), HFD did not alter body weight, liver weight, or the ratio of liver weight to body weight in HuR cKO mice (Fig. 3a). Even under HFD condition, food uptake was comparable between HuR cKO mice and WT mice (Fig. 3a). Importantly, however, HuR cKO mice exhibited much more severe NAFLD than their WT littermates, as evidenced by assessing H&E and Oil Red O staining (Fig. 3b). Similarly, the levels of liver triglycerides and cholesterol in HuR cKO mice were much higher than those in WT mice (Fig. 3c), and deletion of HuR reduced the levels of ATP and mtDNA (Fig. 3d). In the HFD-induced NAFLD mouse model, deletion of HuR reduced the levels of triglyceride, cholesterol, APOB, APOE, HDL-C, and LDL-C in serum (Fig. 4a), while it elevated the serum levels of ALT, AST, and T-Bil, and reduced the levels of ALB and TP (Fig. 4b); it also reduced the glucose tolerance and insulin tolerance (Fig. 4c, d). By liver and serum lipid and metabolomic analysis, the changes of liver and serum lipid and energy metabolites tended to be consistent with the data shown in Fig. 3c, d and Fig. 4a (the list of the changed metabolites was shown in Supplementary Figs. 5–9). In addition, deletion of HuR elevated the levels of ROS and the protein levels of γH2AX in response to HFD (Supplementary Fig. 10). These results support the view that HuR is important for maintaining serum lipid levels and liver function in HFD condition.

The aggravation of HFD-induced NAFLD by HuR deficiency was also observed in mice fed HFD for 12 weeks (Supplementary Fig. 11). As anticipated, the levels of APOB-100, APOB-48, APOE, CYCS, NDUFB6, and UQCRB in HuR cKO mice fed HFD were significantly lower than those observed in WT mice (Fig. 5a, b). In keeping with the findings in Fig. 2c, HuR cKO mice feeding with HFD exhibited reduced levels of *Apob* and *Apoe* mRNAs, but not *Cycs*, *Ndufb6*, or *Uqcrb* mRNAs (Fig. 5c). In C57BL/6

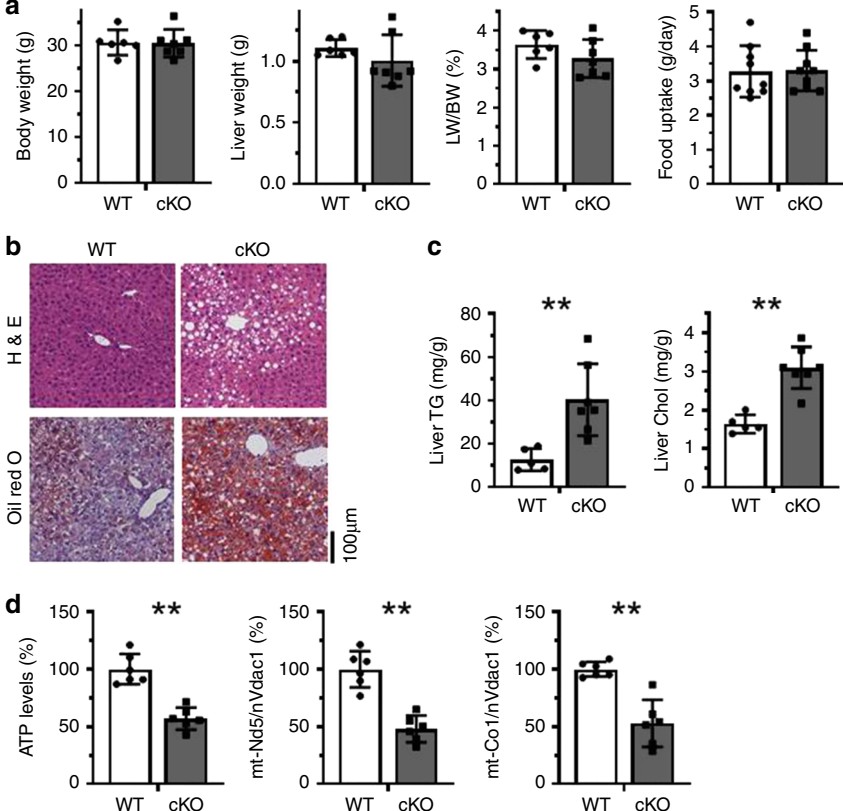

**Fig. 3 Liver HuR deficiency aggravates HFD-induced NAFLD. a** HuR cKO and WT littermates were fed HFD for 4 weeks, whereupon body weight, liver weight, and ratio of liver/body weight were measured (WT, $n = 6$; cKO, $n = 7$). The food uptake was also measured (WT and cKO, $n = 9$). Data are the means ± SD (Blank columns, WT; Black columns, cKO). **b** Representative images of liver sections staining by using H&E and Oil Red O to evaluate the NAFLD phenotype in mice described in **a** (WT, $n = 5$; cKO, $n = 5$). **c** Livers of mice described in **a** were used for analyzing the levels of triglyceride (TG) and cholesterol (Chol) (WT, $n = 5$; cKO, $n = 7$). **d** Mouse livers described in **a** were used for analyzing the levels of ATP and the relative levels of mtDNA (WT, $n = 6$; cKO, $n = 6$). Data in **c** and **d** are the means ± SD; significance was analyzed by using two-tailed Mann–Whitney $U$ test (*$p < 0.05$; **$p < 0.01$). All the error bars are equivalent throughout the figure. Source data are provided as a Source Data file.

mice fed HFD, the protein levels of HuR, APOB-100, APOB-48, APOE, CYCS, NDUFB6, and UQCRB were increased (Supplementary Fig. 12). Accordingly, the protein levels of HuR in both cytoplasm and nucleus were also increased in response to HFD (Supplementary Fig. 13). We therefore propose that HuR promotes lipid transport and ATP synthesis by regulating the expression of APOB-100, APOB-48, APOE, CYCS, NDUFB6, and UQCRB.

**HuR binds *Apob* pre-mRNA and regulates *Apob* mRNA splicing.** In light of the finding that HuR ablation reduced the levels of proteins APOB-100, APOB-48, and APOE, as well as the levels of *Apob* and *Apoe* mRNAs in mice fed normal diet and HFD (Figs. 2, 5), we asked if HuR regulates APOB and APOE expression at the post-transcriptional level. To this end, we performed RNA pull-down assays by using in-vitro-transcribed, biotinylated fragments of *Apob* and *Apoe* mRNAs (Supplementary Fig. 14, Schematic). As shown (Fig. 6a, Supplementary Fig. 15a), HuR did not appear to associate with any of the fragments of *Apob* and *Apoe* mRNAs, suggesting that HuR does not directly regulate *Apob* or *Apoe* mRNAs and hence likely does not modulate their turnover or translation.

Given that HuR can also participate in pre-mRNA processing[26], bioinformatic prediction was used[27] to identify putative HuR-binding sites in introns 5, 8, 10, 12, 14, 24, and 29 of *Apob* pre-mRNA. RNA pull-down assays confirmed that HuR was capable of associating with *Apob* pre-mRNA introns 10, 12, 14,

and 24, but not introns 5, 8, or 29 (Fig. 6a; Supplementary Fig. 14, Schematic). HuR did not associate with any of the introns of *Apoe* pre-mRNA (Supplementary Fig. 15b), suggesting that the reduction of APOE expression in HuR cKO mice (Figs. 2a, b, 5a, b) was an indirect effect from an unknown regulatory process.

Next, we tested if HuR regulates *Apob* mRNA splicing by associating with the introns of *Apob* pre-mRNA. Given that HuR showed stronger interaction with introns 10 and 24 (Fig. 6a), we constructed reporter vectors containing these introns; a reporter vector containing intron 5 served as a negative control (Fig. 6b, Schematic). The levels of the pre-RNAs (Pre) expressed from these reporters and the mature RNAs (RNA) processed from these pre-RNAs were analyzed in cells expressing normal levels of HuR or expressing lower levels of HuR by siRNA-mediated silencing. As shown in Fig. 6c, HuR silencing significantly elevated pre-RNA levels expressed from the reporter containing intron 24, along with a reduction of the processed products (RNA) of this pre-RNA. However, the levels of pre-RNAs and RNAs from the reporters for introns 5 and 10 in cells silenced with HuR were comparable to those observed in control cells. HuR silencing also significantly reduced the levels of endogenous *Apob* mRNA levels and increased *Apob* pre-mRNA levels (Fig. 6d). Likewise, while HuR ablation was earlier shown to lower *Apob* mRNA levels (Fig. 2c, Fig. 5c), it increased the levels of *Apob* pre-mRNA in both normal chow diet and HFD (Fig. 6e). Interestingly, HFD induced the levels of *Apob* pre-mRNA in both

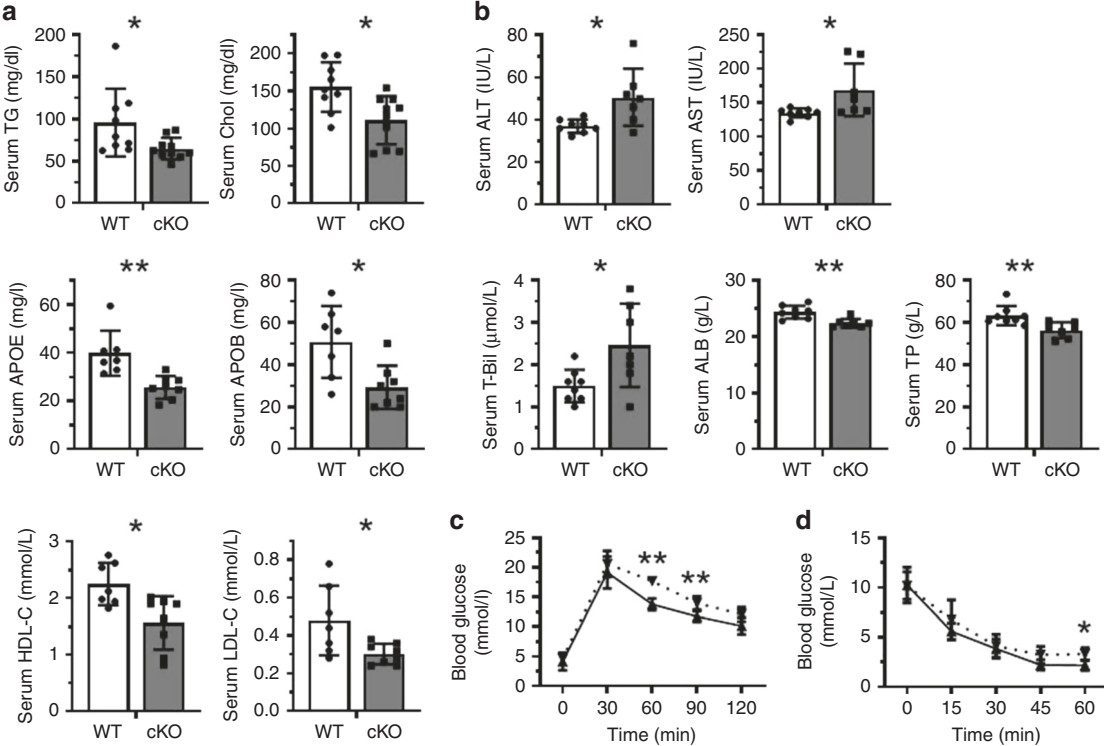

**Fig. 4 Liver HuR deficiency impairs liver function in HFD mice. a** HuR cKO mice and WT littermates were fed with HFD for 4 weeks, whereupon the levels of triglyceride (TG), cholesterol (Chol), APOB, APOE, high-density lipoprotein cholesterol (HDL-C), and low-density lipoprotein cholesterol (LDL-C) in the serum were analyzed (TG and Chol, WT, $n = 9$, cKO, $n = 10$; others, WT, $n = 7$, cKO, $n = 8$). **b** The serum levels of alanine aminotransferase (ALT), aspartate aminotransferase (AST), total bilirubin (T-Bil), albumin (ALB), and total protein (TP) were analyzed (WT, $n = 8$; cKO, $n = 7$). **c** Mice described in **a** were starved overnight (WT, $n = 6$; cKO, $n = 6$) and injected intraperitoneally with glucose (1 g/kg), and serum collected from tail vein at the times indicated were used to analyze the levels of glucose (glucose tolerance test, GTT) by using a glucometer (BAYER). Dotted line, HuR cKO; solid line, WT. **d** Mice described in **a** (WT, $n = 5$; cKO, $n = 7$) were starved for 6 h and injected intraperitoneally with insulin (0.85 U/kg). The levels of glucose in the serum (insulin tolerance test, ITT) were analyzed as described in **c**. Dotted line, HuR cKO; solid line, WT. Data in **a**–**d** are the means ± SD; significance was analyzed by two-tailed Mann–Whitney $U$ test (*$p < 0.05$; **$p < 0.01$). All the error bars are equivalent throughout the figure. Source data are provided as a Source Data file.

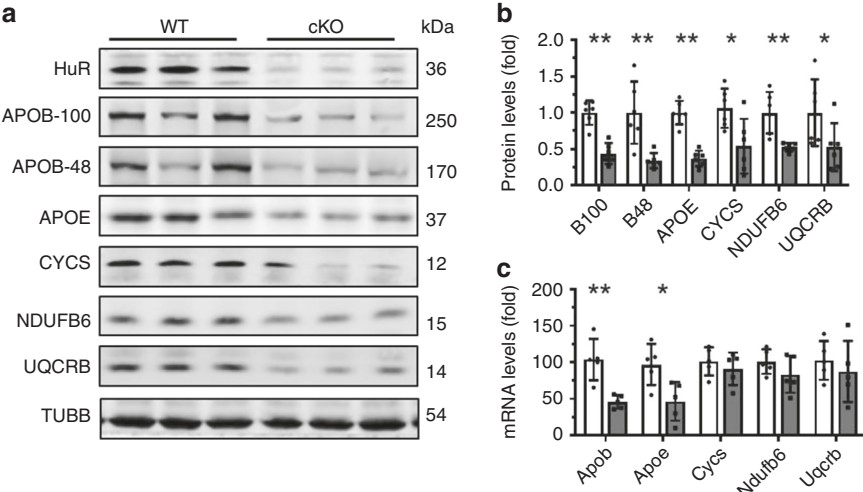

**Fig. 5 Liver HuR ablation reduces the levels of factors regulating lipid transport and ATP synthesis in HFD-fed mice. a** Protein lysates prepared from the mouse livers (WT and cKO, $n = 6$) described in Fig. 3 were subjected to western blot analysis to assess the levels of proteins HuR, APOB-100, APOB-48, APOE, CYCS, NDUFB6, UQCRB, and TUBB. Blots were processed from parallel gels. **b** The density of the protein signals for APOB-100(B100), APOB-48 (B48), APOE, CYCS, NDUFB6, and UQCRB is represented as the means ± SD (WT and cKO, $n = 6$); significance is analyzed by two-tailed Mann–Whitney $U$ test (*$p < 0.05$; **$p < 0.01$). Blank columns, WT; Black columns, cKO. **c** RNA prepared from livers described in **a** was subjected to RT-qPCR analysis to assess the levels of *Apob*, *Apoe*, *Cycs*, *Ndufb6*, and *Uqcrb* mRNAs (WT and cKO, $n = 5$). Data are the means ± SD (Blank columns, WT; Black columns, cKO); significance was assessed by using two-tailed Mann–Whitney $U$ test (*$p < 0.05$; **$p < 0.01$). All the error bars are equivalent throughout the figure. Source data are provided as a Source Data file.

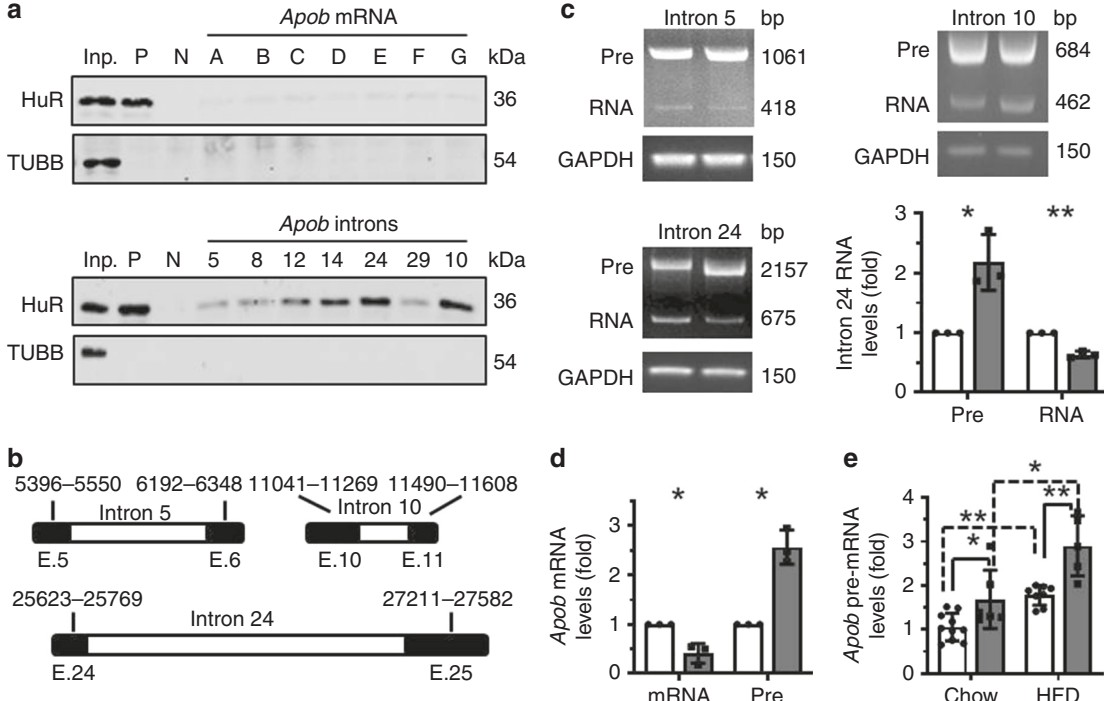

**Fig. 6 HuR regulates *Apob* pre-mRNA splicing. a** RNA pull-down assays were performed using Hepa1–6 cell lysates and in-vitro-transcribed RNAs depicted in Supplementary Fig. 14. *p27* 5′UTR and CR (coding region) served as positive (P) and negative (N) controls, respectively. A 5-µg aliquot input (Inp.) and binding to TUBB were also assessed. Data are representative from three independent experiments. **b** Schematic presentation depicting the RNA fragments inserted into the pcDNA3.1 minigene reporters. **c** The splicing of the pre-RNA expressed from the mini reporters described in **b** was analyzed in Hepa1–6 cells with silenced HuR, as described (Methods). The pre-RNA (Pre) and mature RNA (RNA) were measured by semi-quantitative PCR analysis. Blank columns, control; Black columns, siHuR. **d** Hepa1–6 cells were transfected with a HuR siRNA; 48 h later, RNA was prepared and subjected to RT-qPCR analysis to analyze the levels of *Apob* mRNA (mRNA) and pre-mRNA (Pre). Blank columns, control; Black columns, siHuR. **e** RNA described in Fig. 2c and Fig. 5c was used to analyze the levels of *Apob* mRNA pre-mRNA (Blank columns, WT; Black columns, cKO). Data in **c** (for intron 24) and **d** are the means ± SD from three independent experiments; Data in **e** are the means ± SD (Chow: WT, $n = 10$, cKO, $n = 6$; HFD: WT, $n = 8$, cKO, $n = 5$); significance is analyzed by two-tailed Student's *t*-test (**c**, **d**) or two-tailed Mann–Whitney *U* test (**e**) (*$p < 0.05$; **$p < 0.01$). All the error bars are equivalent throughout the figure. Source data are provided as a Source Data file.

HuR cKO and WT mice (Fig. 6e), suggesting the involvement of additional regulatory processes that were active in HFD-induced NAFLD. In sum, HuR promotes the splicing of *Apob* pre-mRNA through binding to intron 24. We propose that this regulatory mechanism contributes, at least in part, to the elevation of APOB levels in HFD-induced NAFLD.

**HuR promotes ATP synthesis by regulating CYCS, NDUFB6, and UQCRB**. By enhancing the production of CYCS and COQ7[23,24], HuR promotes ATP synthesis in human cells. Based on the findings that HuR deletion reduced liver ATP levels and lowered production of CYCS, NDUFB6, and UQCRB, but not COQ7, in both normal chow diet and HFD conditions (Figs. 1d, 2a, b, 3d, 5a, b), we hypothesized that HuR might promote ATP synthesis in mouse hepatocytes by controlling the production of CYCS, NDUFB6, and UQCRB. To test this possibility, we measured the levels of ATP and the abundance of proteins CYCS, NDUFB6, and UQCRB in Hepa1–6 cells (mouse hepatoma cells). Lowering HuR reduced the abundance of CYCS, NDUFB6, and UQCRB, but not COQ7 (Fig. 7a). The levels of *Cycs*, *Ndufb6*, and *Uqcrb* mRNAs were not altered in cells with silenced HuR (Fig. 7b), in keeping with the results shown in Figs. 2c, 5c. In agreement with the results shown in Fig. 3d and previous studies[23,24], HuR knockdown reduced cellular ATP levels (Fig. 7c). To further investigate the mechanisms underlying the regulation of CYCS, NDUFB6, and UQCRB by HuR,

in-vitro-transcribed, biotinylated RNA fragments of the 5′UTR, CR (coding region), and 3′UTR of *Cycs*, *Uqcrb*, and *Ndufb6* mRNAs (Supplementary Fig. 16a–c, Schematic) were used for RNA pull-down assays. As shown in Fig. 7d, HuR associated with *Cycs* 3′UTR (3′2), with *Ndufb6* 5′UTR, and with *Uqcrb* 3′UTR, but not with other regions of these mRNAs.

To test if the association of HuR with *Cycs*, *Ndufb6*, and *Uqcrb* mRNAs was functional, pGL3-derived reporters bearing fragments of *Cycs*, *Ndufb6*, and *Uqcrb* mRNAs were constructed (Supplementary Fig. 16d, Schematic). Hepa1–6 cells were transfected with each of these reporters and 24 h later, they were transfected with siRNAs (control or HuR-directed) and cultured for an additional 48 h. The ratios of firefly luciferase (FL) activity against Renilla luciferase (RL) activity declined after silencing HuR in pGL3-derived reporters bearing *Cycs* 3′2, *Ndufb6* 5′UTR, and *Uqcrb* 3′UTR, but not when assaying reporters bearing regions that did not interact with HuR. Together, our findings indicate that HuR associates with mRNAs encoding *Cycs*, *Ndufb6*, and *Uqcrb*, promoting their production and enhancing ATP synthesis.

**Modulating of hepatic HuR influences HFD-induced NAFLD**. To further investigate the role of HuR in HFD-induced NAFLD, HuR cKO mice were injected with an adeno-associated virus (AAV) expressing HuR (cKO-HuR) or GFP (cKO-GFP). After 4 weeks of HFD, the levels of APOB-100, APOB-48, CYCS, NDUFB6, and UQCRB were analyzed by western blot analysis.

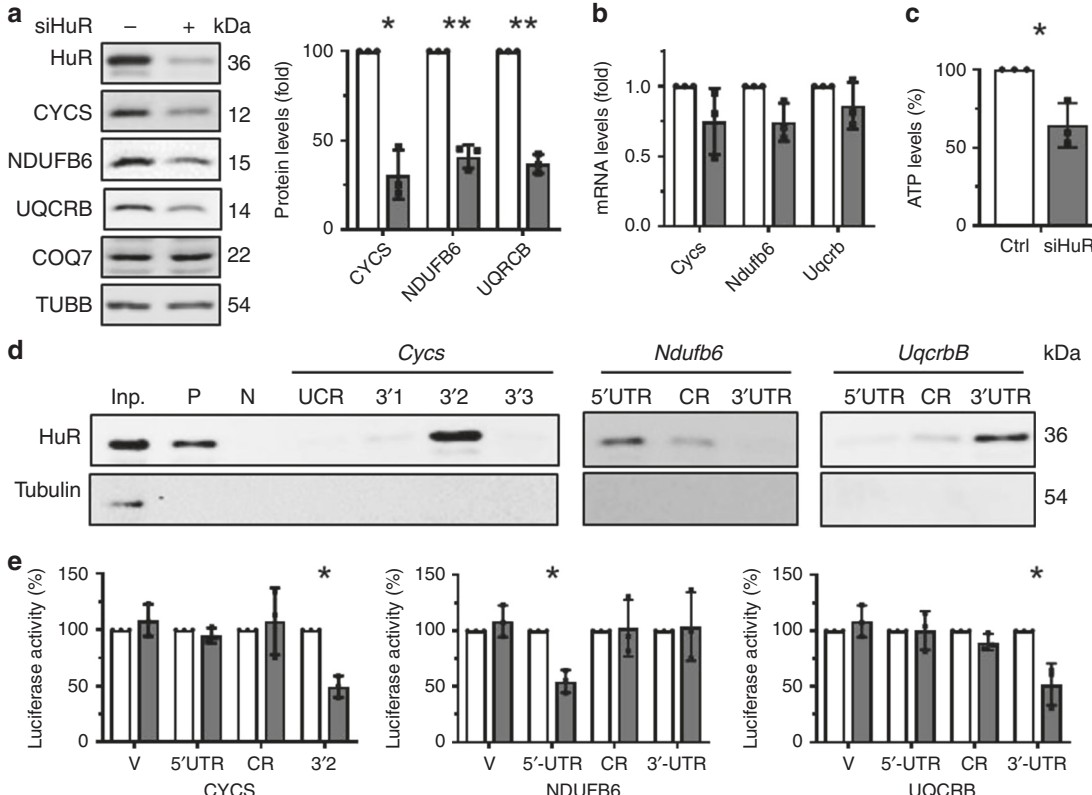

**Fig. 7 HuR promotes ATP synthesis at least in part by regulating the levels of CYCS, NDUFB6, and UQCRB. a**, **b** Mouse Hepa1–6 cells were transfected with HuR-directed or control siRNAs; 48 h later, protein lysates and RNA prepared from cells were subjected to western blot analysis (**a**) and RT-qPCR analysis (**b**), respectively, to assess the levels of HuR, CYCS, NDUFB6, COQ7, UQCRB, and TUBB proteins (**a**) as well as the levels of *Cycs*, *Ndufb6*, and *Uqcrb* mRNAs (**b**), respectively. Blank columns, control; Black columns, siHuR. **c** Cells described in **a** were subjected to ATP analysis. **d** RNA pull-down assays were performed by using Hepa1–6 cell lysates and in-vitro-transcribed RNAs depicted in Supplementary Fig. 16a. *p27* 5′UTR and CR (coding region) served as positive (P) and negative (N) controls, respectively. A 5-μg aliquot input (Inp.) and binding to TUBB were also assessed. Data are representative from three independent experiments. **e** Hepa1–6 cells were transfected individually with each of the reporters depicted in Supplementary Fig. 16d. Twenty four hour later, cells were further transfected with a HuR siRNA or a control siRNA and cultured for an additional 48 h, whereupon the relative luciferase activities were determined. Blank columns, control; Black columns, siHuR. Data in panels (**a**–**c** and **e**) are the means ± SD from three independent experiments; significance is analyzed by two-tailed Student's *t*-test (\*\**p* < 0.01). All the error bars are equivalent throughout the figure. Source data are provided as a Source Data file.

As shown, HuR re-expression increased the levels of APOB-100, APOB-48, CYCS, NDUFB6, and UQCRB proteins (Fig. 8a–b), and reduced *Apob* pre-mRNA levels while it increased *Apob* mRNA levels (Fig. 8c). Importantly, this intervention improved markedly the NAFLD phenotype, as evidenced by H&E and Oil Red O staining (Fig. 8d), and the accumulation of liver triglyceride and cholesterol was also greatly mitigated (Fig. 8e). Because HuR is an RNA-binding protein with multiple target RNAs, we also tried to rescue the effect of HuR in HFD-induced NAFLD by using an AAV that expressed the 3′UTR fragment of *Uqcrb* mRNA, which could compete with the endogenous target mRNAs of HuR to associate with HuR. As anticipated, expression of the 3′UTR fragment of *Uqcrb* mRNA (Supplementary Fig. 17) ameliorated the effect of HFD in inducing the levels of proteins HuR, APOB-100, APOB-48, CYCS, NDUFB6, and UQCRB (Supplementary Fig. 18a–b), but it enhanced the HFD-induced NAFLD phenotype (Supplementary Fig. 18c–d). Of note, the induction of HuR by HFD was also rescued by expression of the *Uqcrb* 3′UTR (Supplementary Fig. 18a), since HuR was found to associate with the 3′UTR of *HuR* mRNA, thereby promoting the nuclear export of the *HuR* mRNA[28]. These results support the view that HuR mitigates HFD-induced NAFLD by increasing the expression of factors involved in lipid transport and ATP synthesis. However, overexpression of CYCS in HuR cKO mice

by infection with an AAV that expressed CYCS did not rescue the effect of HuR deletion in aggravating HFD-induced NAFLD (Supplementary Fig. 19). Although ATP production was mildly elevated by CYCS overexpression, this elevation was not significant (Supplementary Fig. 19). These findings indicate that overexpression of a single target of HuR (e.g., CYCS) may be insufficient to rescue effect of HuR cKO in influencing NAFLD pathogenesis.

## Discussion

The results of this investigation indicate that HuR prevents HFD-induced NAFLD in mice by regulating lipid transport and ATP synthesis (Supplementary Model 1). Mechanistically, HuR elicits this function by associating with *Cycs*, *Ndufb6*, and *Uqcrb* mRNAs as well as *Apob* pre-mRNA, and thereby promoting the expression of CYCS, NDUFB6, UQCRB, and APOB. HuR mediates these associations by binding different segments of precursor and mature mRNAs: the 3′UTRs of *Cycs* and *Uqcrb* mRNAs, the 5′UTR of *Ndufb6*, and intron 24 of *Apob* pre-mRNA. These regulatory processes may also be important for regulating the abundance of CYCS, NDUFB6, UQCRB, and APOB under caloric restriction (CR) diet and metformin administration, since CR diet and oral metformin intake induced the levels of HuR, CYCS, NDUFB6, UQCRB, APOB-100, and APOB-48

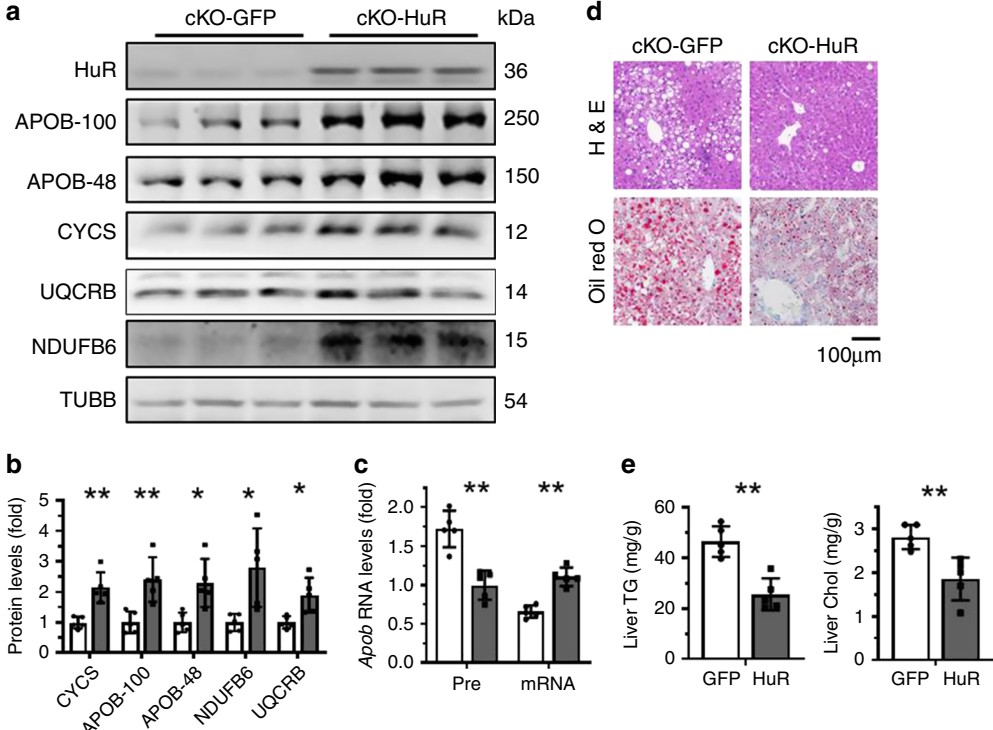

**Fig. 8 Restoring HuR levels rescues HFD-induced NAFLD in HuR cKO mice. a–c** HFD-fed HuR cKO mice were injected simultaneously with an adenovirus expressing HuR (cKO-HuR) or GFP (cKO-GFP). Four weeks later, liver protein lysates and RNA were prepared and subjected to western blot analysis (cKO-HuR and cKO-GFP, $n = 5$) and real-time qPCR (cKO-HuR and cKO-GFP, $n = 5$) to determine the levels of proteins APOB-100 (B-100), APOB-48 (B48), CYCS, NDUFB6, UQCRB, and TUBB (**a, b**), as well as the levels of *Apob* mRNA (mRNA) and pre-mRNA (pre) (**c**), respectively. Western blots in **a** were processed from parallel gels. Data of intensity of the western blots in **a** and the RT-qPCR are presented as the means ± SD in panels **b** and **c**, respectively; significance was analyzed by two-tailed Mann–Whitney $U$ test (**$p < 0.01$). Blank columns, GFP; Black columns, HuR. **d** Representative images of staining liver sections from mice described in **a** using H&E and Oil Red O (cKO-HuR and cKO-GFP, $n = 5$). **e** The levels of liver triglyceride (TG) and cholesterol (Chol) were determined. Data are the means ± SD (cKO-HuR and cKO-GFP, $n = 5$; significance is analyzed by two-tailed Mann–Whitney $U$ test (**$p < 0.01$). All the error bars are equivalent throughout the figure. Source data are provided as a Source Data file.

(Supplementary Fig. 20). In addition, liver tissues from NAFLD patients exhibited higher protein levels of HuR, CYCS, NDUFB6, UQCRB, and APOB, supporting the view that HuR may be important for regulating the production of CYCS, NDUFB6, UQCRB, and APOB during the process of human NAFLD (Supplementary Fig. 21a). Furthermore, HuR likely regulates APOB production in human cells in a similar fashion, given that *APOB* pre-mRNA levels decreased and *APOB* mRNA levels increased in the liver of NAFLD patients, although the differences did not reach significance (Supplementary Fig. 21b). Additional evidence supporting the notion that HuR might regulate human *APOB* mRNA splicing is presented in Supplementary Fig. 22, where the results showed that HuR associated with intron 24 of human *APOB* pre-mRNA (Supplementary Fig. 22a) and silencing HuR in human hepatoma cells (HepG2) inhibited *APOB* mRNA splicing (Supplementary Fig. 22b).

Although HuR does not associate with *Apoe* mRNA or pre-mRNA, the reduction of APOE levels in HuR cKO mice consuming normal chow diet or HFD suggests that HuR may regulate APOE expression in vivo. The enhancement of APOE production by HuR may occur via elevated transcription of the *Apoe* gene or via stabilization of *Apoe* mRNA; identifying the mechanisms responsible will require further study.

By a unique mRNA editing mechanism, *Apob* mRNA gives rise to two forms of APOB, APOB-48, and APOB-100[29]. Unlike APOB-100, APOB-48 is predominantly expressed in enterocytes and is obligatory for the packing of lipids into chylomicrons

(CMs), which transport lipids after absorption through the intestinal wall[29]. The regulation of APOB-48 by HuR (Figs. 2, 5) may also underscore a critical role of HuR in the formation of CMs and the consequent transport of the absorbed lipids into peripheral tissues in the heart, skeletal muscle, and adipose compartment.

Although HuR cKO mice exhibited accelerated NAFLD, we did not observe significant alterations in liver weight or the ratio of liver weight against body weight (Figs. 1a and 3a, and Supplementary Fig. 11a). This result may be due to the pleiotropic roles of liver HuR in regulating both cell growth[20,22] and lipid balance, as investigated here. In other words, the effect of HuR deficiency in liver weight contributing from accumulating lipids may be countered by the fact that loss of HuR reduces cell growth.

A recent study reports that ablation of HuR in endothelial cells reduces the expression of proatherogenic molecules and attenuates atherosclerosis[30]. Notably, in HFD-fed HuR cKO mice, the accumulation of liver lipids was accompanied by a reduction in the levels of serum lipids (Fig. 3, supplementary Figs. 5–8). These observations indicate that HuR may further protect arteries from atherosclerosis by helping to maintain lipid homeostasis. A study underway in our laboratory that focuses on the impact of liver HuR deletion in the progression of vascular disorders (including vascular inflammation and atherosclerosis) may illuminate molecular details of the metabolic interaction of the liver and the vasculature. Apart from regulating *Apob* mRNA splicing in human cells (Supplementary Fig. 22), HuR plays important roles

in the processes of liver differentiation, development, and human hepatocellular carcinoma (HCC) as well as liver fibrogenesis[31,32]. Thus, the roles of HuR in human liver diseases (e.g., NAFLD, liver fibrosis, liver cirrhosis, etc.) warrant further study.

In closing, the roles of HuR in tumorigenesis and cardiovascular function are well documented[33–35]. These processes are characterized by dysregulated mitochondria function and ATP synthesis, and by imbalances in lipid metabolism[36–40]. Therefore, we postulate that HuR may also impact upon these processes by controlling the production of CYCS, NDUFB6, UQCRB, and APOB, in turn preserving the ability of mitochondria to produce energy and maintaining lipid homeostasis.

## Methods

**Ethics statement**. The animal facility was accredited by the AAALAC (Association for Assessment and Accreditation of Laboratory Animal Care International). All mouse husbandry and experiments were carried out in accordance with the Guide for the Care and Use of Laboratory Animals of the Health Science Center of Peking University, and the IACUC (Institutional Animal Care and Use Committee) of Model Animal Research Center of Nanjing University. Mice were housed in groups with 12-h dark-light cycles and had free access to food and water.

**Mice and high-fat diet**. *Elavl1* (*HuR*) floxed mice were purchased from the Jackson Laboratories[41]. Hepatocyte-specific *Elavl1* deletion mice were generated by crossing *Elavl1*floxed mice with *Albumin* (*Alb*)-Cre mice. All mouse lines were maintained in a C57BL/6 J genetic background. Mice were housed in groups of 1–6 with 12-h dark-light cycles at room temperature (~24 °C), and had free access to food and water. To induce NAFLD, 8-week-old *HuR* Flox/Flox male mice and their WT littermates were fed high-fat diet (HDF; Research Diets D12492: 60% Kcal fat, 20% Kcal protein, and 20% Kcal sucrose) for 4 weeks or 12 weeks. Mice then were sacrificed under isoflurane inhalation (1.4%) followed by cervical dislocation.

**Cell culture and transfection**. Mouse hepatoma Hepa1–6 cells were cultured in Dulbecco's modified Eagle's medium (Invitrogen) supplemented with 10% fetal bovine serum and antibiotics, at 37 °C in 5% $CO_2$. All plasmid transfections were performed using lipofectamine 2000 (Invitrogen). Unless otherwise indicated, cells were analyzed 48 h after transfection. To silence HuR transiently, cells were transfected with an siRNA targeting HuR (UUGUUCGAACGU GUCACGUUU) or a control siHuR (AAGAGGCAAUUACCAGUUUCA) by using Lipofectamine RNAiMAX (Invitrogen).

**Infection of adeno-associated virus (AAV) in mouse hepatocytes**. Adeno-associated viruses expressing HuR or GFP (Hanbio Biotechnology, Cat# HH2018 0129FF-AD01), CYCS (Hanbio Biotechnology, Cat# HH20190702YT-AAV02), as well as that expressing *Uqcrb* CR (GATCTGCTGTTTCAGCATCAAGCAAGTGG CTGGATGGTTTTCGAAAGTGGTATTATAATGCTGCAGGATTCAATAAAC TGGGGTTAATGCGAGATGATACACTACATGAAACTGAAGATGTAAAAGA AGCCATAAGAAGGCTTCCTGAGGACCTTTATAATGACAGGATGTTTCGA ATTAAGAGAGCCCTGGACCTGACTATGAGGCATCAGATCTTGCCTAAGG ATCAGTGGACAAAATATGAGGAGGACAAATTCTACCTTGAACCCTATCT AAAAGAGGTTATTCGGGAAAGAAAGGAGAGAAAGAGTGGGCAAAGAA GTG) (Hanbio Biotechnology, Cat# HH20190216HR-AAV01) or 3′UTR (GAAG AGTGGGCAAAGAAGTGATCTTTTAGTTAAGATCTGTGGGTGTGCCTGGT CTCACCATACTTTTACAAAGTTATTTCAACCCAAATCACAATTTAAGAAT TATTTGTTCTACCTATGCCACACTTTAAATAAATGTCTATTATAAC) (Hanbio Biotechnology, Cat# HH20190216HR-AAV02) were injected from the tail vein of HuR cKO mice (~1 × 10¹¹vg/mouse). The efficiency of virus infection and the expression of HuR in mouse hepatocytes were further confirmed by Western blot analysis.

**Antibodies and western blot analysis**. Western blot analysis was performed using standard procedures. Polyclonal anti-HuR, polyclonal anti-APOB, polyclonal anti-Cd36, polyclonal anti-FABP1, polyclonal anti-PPARα, polyclonal anti-CPT1, polyclonal anti-NDUFB6, polyclonal anti-UQCRB, polyclonal anti-COQ7, polyclonal anti-ATP6V1α, and polyclonal anti-TUBB were from Proteintech. Monoclonal anti-CYCS and anti-γH2AX were from Abcam. Polyclonal anti-APOE, polyclonal anti-MTTP, and polyclonal anti-SCD1 were from affinity.

**Reporter activities**. Transient transfection of Hepa1–6 cells cultures with the reporters was carried out by Lipofectamine 2000 (Invitrogen). Cotransfection of pRL-CMV served as an internal control. Firefly and renilla luciferase activities were measured with a double luciferase assay system (Promega, Madison, Wis.) following the manufacturers' instructions. All firefly luciferase measurements were normalized to renilla luciferase measurements from the same sample.

**Measurement of the ATP levels and mtDNA copies**. For measuring the levels of ATP, adenosine 5′-triphosphate (ATP) Bioluminescent Assay Kit (Sigma) was used to test ATP, following the manufacturer's instructions. To measure the copies of mitochondria, genomic DNA was extracted from liver tissue or cells by TaKaR-aMiniBEST Universal Genomic DNA Extraction Kit (Code No.9765). For PCR analysis of the relative mitochondria DNA content, we used the following primer pairs: ACATGAAACCCCCAGCCATA and TTGTGTTTAGGTTGCGGTCTG for *mt-Co1*, TGCCTAGTAATCGGAAGCCTCGC and TCAGGCGTTGGTGTTGCA GG for *mt-Nd5*, and CTCCCACATACGCCGATCTT and GCCGTAGCCCTTG GTGAAG for *Vdac1*, as reported[42]. MtDNA copy number was calculated relative to nuclear DNA using the following procedure[43].

$$\Delta C_T = \text{mitochondrial } C_T - \text{nuclear } C_T \quad (1)$$

$$\text{Relative mitochondrial DNA content} = 2 \times 2^{-\Delta CT} \quad (2)$$

**Serum and liver biochemical analysis**. Serum TG (total triglyceride), TC (total cholesterol), HDL-C (high-density lipoprotein cholesterol), LDL-C (low-density lipoprotein cholesterol), ALT (alanine aminotransferase), AST (aspartate aminotransferase), T-Bil (total bilirubin), TP (total protein), and ALB (albumin) were determined by using an Autumatic biochemical analyzer in the clinical laboratory of Peking University Third Hospital. Total triglyceride and total cholesterol of liver were extracted as described[44]. Liver tissues were homogenized with a 20-fold excess of 2:1 chloroform–methanol mixture (v/v). The homogenates then were filtered by using fat-free filter paper and analyzed for the levels of total triglyceride and cholesterol by using the same methods as those used to analyze serum triglycerides and cholesterol. Serum APOB and APOE were measured by using an ELISA Kit (Cloud-Clone Corporation) following the manufacturer's instructions.

**Analysis of GTT, ITT, and metabolic cage parameters**. Glucose tolerance tests (GTT) and insulin tolerance tests (ITT) were performed after fasting mice for 12 h or 6 h, respectively, and then injecting glucose (1 g/kg) or insulin (0.85U/kg) intraperitoneally. Serum samples were collected from the tail vein and the serum glucose levels were tested by using a glucometer (BAYER). For collecting metabolic cage parameters, HuR cKO mice were fed with HFD for 1 month; mice then were housed individually in metabolic cages at a 12-h light-dark cycles using the CLAMS (Columbus Instruments) (equipped with metabolic cage analyzer). Mice were acclimated to the metabolic cage for 1 day before recording data. The metabolic cage collected the velocity of oxygen consumption ($VO_2$), $CO_2$ production (VCO2), and energy expenditure (EE) over several days.

**RNA isolation, reverse transcription (RT) and quantitative real-time (q)PCR analysis**. RNA was extracted from livers with Trizol (Tiangen) and treated with Superscript II reverse transcriptase (RT) and the mixture of oligo-dT and random hexamers (Invitrogen). A two-step real-time qPCR amplification reaction was performed by using a SYBR green (Enzyme) and a Bio-Rad CFX96TM Real-time System. To quantify the levels of different mouse mRNAs, RT-qPCR analysis was carried out using the following primer pairs:
GGAGCCATGGATTGCACATT and GCTTCCAGAGAGGGAGGCCAG for *Srebp1c* mRNA, GAAAAGCAGTTCAACGAGAACG and AGATGCCGACCAC CAAAGATA for *Elovl6* mRNA, ATGGGCGGAATGGTCTCTTTC and TGGGGA CCTTGTCTTCATCAT for *Acaca* mRNA, AAGTTGCCCGAGTCAGAGAACC and ATCCATAGAGCCCAGCCTTCCATC for *Fasn* mRNA, TGGTGACTTTAT GGAGCCTAA and GGCGAACAGCTGAGAGGACTCTG for *Pparγ* mRNA, CGTGGGCTCCAGCATTCTA and TCACCAGTCATTTCTGCCTTTG for *Apob* mRNA, GCTGGGTGCAGACGCTTT and TGCCGTCAGTTCTTGTGTGACT for *Apoe* mRNA, GGGCACGGCTATTCTCACAG and CATCAAGAACCTGGCCG TCT for *Acox1* mRNA, CGGTCAATGCCATCAGTCCAA and TGCTCCACA GATCACTATGGC for *Ehhadh* mRNA, GTGGAAAGTAGACCGGAACGA and CCATCCTGTGTGATTGTCAGTT for *Fabp2* mRNA, ACCAAGCCTACTACCA TCATCG and CCTCGTCGAACTCTATTCCCAG for Fabp3 mRNA, TGGACC AAATCTCCACGGTCTGTT and TAGGTCTGCCCTTTCTCCCTTCTT for *Cycs* mRNA, ATAACTTTTTGCGGGACGGG and CAGGAAAATCTCTCATTGGTG for *Ndufb6* mRNA, and TTCAGCATCAAGCAAGTGG and ATCATCTCGCAT TAACCCCAGT for *Uqcrb* mRNA.
Additional primer pairs included for amplification of other transcripts were as follows: ACCAATGGTGTGTAGTACAAAGTC and GTACTTTCGGAGGTGCT TGAAT for mouse *Apob* pre-mRNA, GGGAGCCTGAGAAACGGC and GGGTC GGGGAGTGGGTAATTT for mouse *18 S* rRNA, GCACAAACACTGTGGCATCT and AACGCCTCTAGCCATTCTGA for human *Apob* pre-mRNA, TGACCTTG TCCAGTGAAGTC and GTTCTGAATGTCCAGGGTGA for human *Apob* mRNA, and CAGCCACCCGAGATTGAGCA and TAGTAGCGACGGGCGGT GTG for human *18 S* rRNA.

**Staining with hematoxylin and eosin (H & E) and with Oil red O**. Fresh liver sections were fixed in 20% Formalin and then embedded in paraffin for staining with hematoxylin and eosin (H&E). Fixed liver slices were embedded in optimal cutting temperature compound (OCT) and then stained with Oil Red O (Sigma–Aldrich, St. Louis, MO).

**Constructs**. To construct pGL3-derived reporters bearing the 5′UTR fragments of *Cycs*, *Ndufb6*, and *Uqcrb* mRNAs, the 5′UTR fragments were amplified by PCR using following primer pairs: CCCAAGCTTACGTCTGTCTTCGAGTCCGA and CATGCCATGGTTTTAATTCGTTCCG for *Cycs* 5′UTR, CCCAAGCTTGCGCC GCTTCCAGGCGCC and CATGCCATGGTACCCTGACATGTTGCCG for *Ndufb6* 5′UTR, and CCCAAGCTTGGCTTACCCAGAAGGC and CATGCCAT GGCTTGATGCTGAAACAGCAGAT for *Uqcrb* 5′UTR. These fragments then were inserted between the HindIII and NcoI sites of the pGL3-promoter vector (Promega).

To construct pGL3-derived reporters bearing the CR or 3′UTR of *Cycs*, *Ndufb6*, and *Uqcrb* mRNAs, fragments for insertion were amplified using the following primer pairs: GCTCTAGAATGGGTGATGTTGAAAAAG and GCTCTAGATTA CTCATTAGTAGCC for *Cycs* CR, GCTCTAGACCTCATCGTGACTTCAGACG and GCTCTAGACTTGGGCTCCCAGAAAGGT for *Cycs* 3′UTR2, GCTCTAG ACGGCAACATGTCAGGGTA and GCTCTAGAGGACTCTTCAATGATGTTG ATC for *Ndufb6* CR, GCTCTAGACCTCATCGTGACTTCAGACG and GCTC TAGACAAATCATAGAACCTTTGGACAG for *Ndufb6* 3′UTR, GCTCTAGAGA TCTGCTGTTTCAGCATCAAG and GCTCTAGACACTTCTTTGCCCACTCT TC for *Uqcrb* CR, and GCTCTAGAGAAGAGTGGGCAAAGAAGTG and GCTC TAGAGTTATAATAGACATTTATTTAAAG for *Uqcrb* 3′UTR. These fragments then were inserted into the XbaI sites of the pGL3-promoter vector (Promega).

To construct the minigene reporters to study the splicing of *Apob* pre-mRNA, a fragment containing exon 4 + intron 5 + exon 5 (5396-6348) was generated by PCR by using the primer pair GGGGTACCGTACGAACTCAAGCTGGCC and GCTCTAGACAGGGCCTTTGATGAGAGCGAG and inserting it between the KpnI and XbaI sites of pcDNA3.1 vector (Addgene). Fragment bearing exon 9 + intron 10 + exon 10 (11041-11608) was generated by PCR using primer pairs CGGGATCCCCCCCATCACTTTACAAGCCTTG and GCTCTAGACCTCAG AATCAAGAAGGTGTG and inserted into the BamHI and XbaI sites of the pcDNA3.1 vector. A fragment bearing exon 23 + intron 24 + exon 24 (25623-27582) was generated by using primer pair CGGGATCCGCAACAAACACAT GGCTTCAG and CGGAATTCGCACACTGTAGGAAAACAGG and inserted between the BamHI and EcoRI sites of pcDNA3.1 vector.

**Preparation of transcripts**. cDNA was used as a template for PCR amplification of RNA fragments. All 5′ primers contained the T7 promoter sequence (CCAAGCTTCTAATACGACTCACTATAGGGAGA). To prepare templates for the *Cycs* 5′UTR + CR (11-385), *Cycs* 3′UTR1 (370-1490), *Cycs* 3′UTR2 (1471-2363), *Cycs* 3′UTR3 (2343-3057), *Ndufb6* 5′UTR (1-95), *Ndufb6* CR (78-478), *Ndufb6* 3′UTR (475-626), *Uqcrb* 5′UTR (1-71), *Uqcrb* CR (50-374), and *Uqcrb* 3′ UTR (355-503), the following primer pairs were used: (T7)ACGTCTGTCTTCG AGTCCGA and TTTTAATTCGTTCCG for *Cycs* 5′UTR + CR, (T7)GGCTACT AATGAGAA and CGTCTGAAGTCACGATGAGG for *Cycs* 3′UTR1, (T7)CCTC ATCGTGACTTCAGACG and CTTGGGCTCCCAGCCCCGGTT for *Cycs* 3′ UTR2, (T7)AACCTTTTCTGGGAGCCCAAG and TCCGACATGTCCTTTATT TAGC for *Cycs* 3′UTR3, (T7)GCGCCGCTTCCAGGCGCC and TACCCTGACA TGTTGCCG for *Ndufb6* 5′UTR, (T7)CGGCAACATGTCAGGGTA and GGAC TCTTCAATGATGTTGATC for *Ndufb6* CR, (T7)GATCAACATCATTGAAGAG and CAAATCATAGAACCTTTGGACAG for *Ndufb6* 3′UTR, (T7)GGACTTAC CCAGAAGGC and CTTGATGCTGAAACAGCAGATC for *Uqcrb* 5′UTR, (T7) GATCTGCTGTTTCAGCATCAAG and CACTTCTTTGCCCACTCTTC for *Uqcrb* CR, and (T7)GAAGAGTGGGCAAAGAAGTG and GTTATAATAGACAT TTATTTAAAG for *Uqcrb* 3′UTR.

To prepare templates for the fragments of *Apob* mRNA and pre-mRNAs, the following primer pairs were used: (T7)AAGACCCTGTAGAGCAAGC and TGAT GGCCTGTACAACCATCT for *Apob* A, (T7)AGATGGTTGTACAGGCCATCA and CGCACTGGTTGATAGTCCAT for *Apob* B, (T7)ACGTGATGGACTATC AACCA and TCTGTGGGCCAACTCTACTGA for *Apob* C, (T7)TCAGTAGAGT TGGCCCACAGA and TCACCAGTCATTTCTGCCTTTG for *Apob* D, (T7)CA AAGGCAGAAATGACTGGTGA and TTGCTTTTTAGGGAGCCTAGC for *Apob* E, (T7)GCTAGGCTCCCTAAAAAGCAA and TCAGAAGGTGATCATCAGCT for *Apob* F, (T7)AGCTGATGATCACCTTCTGA and TGCTGCACAATAAGGA AGTA for *Apob* G, (T7)GTTGTTCCTGGTGAGAATC and TGTGGGAGCCCA AGATATG for *Apob* intron 5, (T7)TGCCATGAAATCTATCCACAG and TGGAAGAAGAGAAGTGCCA AG for *Apob* intron 8, (T7)TTAACAGGTGAGACTCCACC and GTCCACATCA AAATAGCTACC for *Apob* intron 10, (T7)GTAAAGTTCTCTGTGTGGTCC and AGACAGAGCATATCTGGTGG for *Apob* intron 12, (T7)GTAAGTGTGACTG AAGGAGC and CTGAGAAAGAGAACCAGGC for *Apob* intron 14, (T7)CGTA AGAGGGGAACTAAAGAG and CTGGAATGAACAAAGACAGTG for *Apob* intron 24, (T7)CCCTAAAAAGCAATGTGCC and GGACCTGTATTTTGAAAA CC for *Apob* intron 29, (T7)TTTCCTCTGCCCTGCTGTGA and AGCCTTCATC TTCGCAATTGTGA for *Apob* 5′UTR, (T7)TCACAATTGCAGAGGTGAAGGCT and CAGGACAGGAGAAGGGATACTCA for *Apob* CR, (T7)TGAGTATCCTTCT CCTGTCCTG and TGGAATCACTCACAGAGACTCA for *Apob* 3′UTR, (T7) GTAAGACAAGCTGGGCTGGGGATTC and CCTAGGGTTCAAATTCCACTT CACAGAGG for *Apoe* intron 1, (T7)GTAAACAGACCTTTGGGA and CTGCG GAGGGAAAAATTG for *Apoe* intron 2, (T7)GTATGGAGCAAGGACTTGCTG TGGTGC and CCTGTGTAGAGTAAGTTTAAGGCTGGGTC for *Apoe* intron3, and (T7)GTGAGTGTCCAGCTCTTTCACCCTC and CTGCAGGCCCAGAGGA

AACACAG for *Apoe* intron 4. The *p27* 5′UTR and CR fragments were described previously[45].

**RNA pull-down**. For biotin pull-down assays, PCR-amplified DNA fragments were used as templates to transcribe biotinylated RNA by using T7 RNA polymerase in the presence of biotin-UTP. One microgram of purified biotinylated transcripts was incubated with 30 μg of cell lysates for 30 min at room temperature. Complexes were isolated with paramagnetic streptavidin-conjugated Dynabeads (Dynal, Oslo), and the pull-down material was analyzed by western blotting.

**RNA splicing analysis**. An *Apob* minigene reporter was transfected into Hepa1–6 cells. Twenty four hour later, cells were further transfected with a HuR siRNA or a control siRNA and cultured for additional 48 h. RNA was isolated and used for semi-quantitative RT-PCR analysis by using primer pairs CMV-Forward (TAAT ACGACTCACTATAGGG) and BGH-reverse (GACTCGCTCCGTTCCTCTTCC TG). The PCR assays were run for 20-25 cycles.

**Isolation of mitochondria**. Cells were collected and suspended in ice-cold IB-1 buffer [225 mM mannitol, 75 mM sucrose, 0.1 mM EGTA and 30 mM Tris-HCl (pH7.4)]. Cells then were homogenized and centrifuged at $600 \times g$ for 5 min at 4 °C. Pellets (unbroken cells and nuclei) were collected and used as negative controls. The supernatants were centrifuged at $600 \times g$ for 5 min at 4 °C once more. The supernatants were further transferred to fresh tubes and centrifuged at $7000 \times g$ for 10 min at 4 °C. The pellets containing mitochondria were resuspended in IB-2 buffer [225 mM mannitol, 75 mM sucrose, and 30 mM Tris-HCl (pH7.4)] and centrifuged again at $7000 \times g$ for 10 min at 4 °C. The pellets were resuspended in IB-2 buffer again and centrifuged at $10,000 \times g$ for 10 min at 4 °C. The crude mitochondrial pellets were resuspended in ice-cold MRB buffer [250 mM mannitol, 5 mM HEPES (pH7.4), and 0.5 mM EGTA] for further analysis.

**ROS measurements**. Reactive oxygen species (ROS) were measured in the primary hepatocytes by using 2′,7′-dichlorodihydrofluorescein diacetate (DCFH-DA) (Nanjing Jiancheng bioengineering Institute, China, #E004) following the manufacturer's instructions. The hepatocytes isolated from the livers of WT and HuR cKO mice were incubated with DCFH-DA at a final concentration of 10 μM at 37 ° C in dark for 40 min. DCF fluorescence intensities in cells was then determined by flow cytometry.

**Immunohistochemistry**. Liver slices from normal human population ($n = 5$) and patients suffering from NAFLD were subjected to immunohistochemistry analysis by Wuhan Servicebio technology co., LTD (Wuhan, China). Polyclonal anti-APOB (ab20737), polyclonal anti-CYCS (ab76107), and polyclonal anti-UQCRB (ab190360) antibodies were from Abcam; polyclonal anti-HuR (11910-1-AP) and polyclonal anti-NDUFB6 (16037-1-AP) antibodies were from Proteintech.

**Caloric restriction diet (CR) and metformin administration**. For caloric restriction, C57BL/6 mice were fed diets in which caloric intake was reduced by 40% (40% CR) for 2 months. For metformin administration, metformin was administered orally to C57BL/6 mice at 300 mg/kg once daily for 3 months.

**Human liver tissue samples**. Human liver samples from patients with NAFLD and healthy control subjects (old samples only be allowed for retrospective study) were provided by Dr. Congxiu Miao and approved by the Research Ethical Committee of Changzhi Medical College (#2019112), informed consent for research and publication was obtained from donors.

**Statistical analysis**. Two-tailed Student's *t*-test was used to analyze the significance of the data obtained from cells. Two-tailed Mann–Whitney *U* test was used to analyze the significance of the data obtained from animals or human samples. Significance was indicated only when p-value < 0.05.

**Reporting summary**. Further information on research design is available in the Nature Research Reporting Summary linked to this article.

## Data availability

The authors declare that all data supporting the findings of this study are available within the article and its Supplementary Information files. The raw data for dot graphs and uncropped versions of any gels or blots or micrographs presented in the figures and supplementary figures are included in the Source Data File. The raw data of lipid and energy metabolomic analysis can be found in the Supplementary Dataset file (Supplementary

Dataset 1). The vectors or mouse models used in this study will be available from the corresponding author upon reasonable request.

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

## Acknowledgements

This work was supported by Grant 2017YFA0504302 from National Key Research and Development Program of China; Grants, 81420108016, 81930035, 81741003, 91749208, 91749203, and 91849108 from the National Natural Science Foundation of China. M.G. was supported by the NIA IRP, NIH.

## Author contributions

Z.Y., Y.D., Z.J., C.J., M.G., de Cabo. R., and W.W. designed the study. Z.Z., C.Z., M.J., H.H., X.C., J.N., X.Y., J.B., T.F., C.M., W.S., L.Z., J.L., X.X., C.M., and J.Y. performed the experiments. M.G., Z.Y., and W.W. wrote the manuscript.

## Competing interests

The authors declare no competing interests.
