## [Peer Review File · Nature Communications]

Reviewers' comments first round:

Reviewer #1 (Remarks to the Author):

Zhang et al put forth a report in studying the role of hepatic HuR in regards to a high fat diet and its relationship to non-alcoholic fatty liver disease. Mechanistically, they found that HuR associated with APOB pre-mRNA (splicing), UQCRB 3'UTR, and NDUFB6 5'UTR (translation). Thus, HuR knockout in the liver dramatically reduced the expression of APOB, UQCRB, and NDUFB6 and functionally reduced lipid transport and ATP synthesis. Rescue experiments were successful. The data presented are strong and the conclusions are adequate. The paper is well presented (minor typos found) and is concise. The work does provide novel insights into this disease and HuR biology. A few major issues should be addressed in order to reach a high impact study:

Mechanism/biology:

- 1) What happens to cytoplasmic/nuclear expression of HuR in these samples/studies (and at different time points-IHC and western)?
- 2) In some of the western blots it doesn't look like HuR is completely knocked out? More detail about the cKO needs to be added to the data and explained. Was qPCR performed as well?
- 3) What about a caloric restriction diet, when the mice get stressed, this may be more relevant in depicting the role of HuR in this context?
- 4) Any metabolomic data available on these tissues?
- 5) What about rescuing with the specific targets to show the importance of these targets?
- 6) Under these conditions what happens to ROS levels and H2AX?
- 7) The phenotype is not striking, have the investigators thought about providing another inducer/stressor to the system beyond modulating the diet?
- 8) The adenoviral rescue work for altering the NAFLD phenotype is not convincing as shown, can the investigators show more convincing/dramatic data?

Human/disease relevant question

- 9) Any more relevant human samples from actual (non-alcoholic fatty liver disease) to show levels of HuR and these targets?

Reviewer #2 (Remarks to the Author):

The goal of this paper was to explore the role of the RNA-binding protein HuR in lipid metabolism and identify potential mechanisms linking HuR and NAFLD. To undertake this goal, the authors created a conditional hepatocyte-specific HuR knockout mouse and evaluated the role of HuR in high-fat diet-induced NAFLD. The major claims of the paper are that HuR associates with key proteins involved in lipid transport, including APOB, UQCRB, and NDUFB6, reduced liver lipid transport, and decreased ATP levels, thereby exacerbating HFD-induced NAFLD.

While HuR has been actively investigated within the context of cancer and aging, its role in lipid metabolism and NAFLD remain unexplored. Therefore, these findings are considered novel and given the results showing a diet-induced effect of HuR, these findings will be of interest to NAFLD researchers. In light of the ubiquitous expression of HuR and its impact on diverse cell activities, the results are also expected to be of interest to the wider community as well.

Injection of HuR cKO mice with adenovirus expressing HuR markedly improved the NAFLD phenotype and elevated levels of APOB-100, APOB-48, CYCS, and NDUFB6, which lends credence

to the major findings. The experimental design is solid and the the work convincing, although it would be interesting to see what effect HuR has on actual progression of NAFLD. Results from this study are likely to influence thinking in the field by underscoring a new candidate for diet-induced NAFLD, as well as a new mechanism by which NAFLD develops in the absence of major disruption in liver function or hepatocyte morphology.

There are no concerns with the statistical analyses.

One suggestion for this paper is to include a schematic depicting the pathway, with all relevant players, by which HuR contributes to pathogenic features of NAFLD.

Point-by-point response letter

Responses to the comments from Reviewer 1:

1) *What happens to cytoplasmic/nuclear expression of HuR in these samples/studies (and at different time points-IHC and western)?*

We appreciate this suggestion. The cytoplasmic/nuclear distribution of HuR in hepatocytes isolated from the livers of mice in the high-fat diet (HFD) group for 1 or 3 months was analyzed by Western blot analysis and IHC. As shown in the supplementary Fig. 8, both cytoplasmic and nuclear HuR were induced by HFD, along with the overall increase in HuR protein levels (Supplementary Figure 7).

IHC assays confirmed that both cytoplasmic and nuclear HuR were increased (Figures 1-2 for reviewer 1). Given that cytoplasmic signals are much lower than nuclear signals, as is the case for HuR in virtually all cell types, we displayed the Western blot analysis (Figure 8 in the revised manuscript). If the reviewer feels that we should include the IHC data, we would gladly do this.

Figure 1 for reviewer 1. C57BL/6 mice were fed with HFD for 4 weeks, whereupon the levels of cytoplasmic and nuclear HuR were tested by immunohistochemistry.

Figure 2 for reviewer 1. C57BL/6 mice were fed with HFD for 12 weeks, whereupon the cytoplasmic and nuclear protein levels of HuR were tested by immunohistochemistry.

2) *In some of the western blots it doesn't look like HuR is completely knocked out? More detail about the cKO needs to be added to the data and explained. Was qPCR performed as well?*

Because the HuR deletion is hepatocyte-specific and the lysates used for Western blot analysis are prepared from whole livers (containing other liver cells in which HuR is not deleted, such as macrophages, endothelial cells, etc), it is expected to see small amounts of HuR. To further confirm the knockout of HuR in hepatocytes, we prepared samples enriched in hepatocytes from the livers of WT and HuR cKO mice, prepared lysates, and assessed HuR levels by Western blot analysis. As shown (supplementary Figure 1c), HuR was not detectable in hepatocytes. This expression pattern is recapitulated for *HuR* mRNA as measured by RT-qPCR analysis (Supplementary Figure 1a).

3) *What about a caloric restriction diet, when the mice get stressed, this may be more relevant in depicting the role of HuR in this context?*

This is another great suggestion by the reviewer. In the revised manuscript, we have included Western blot analysis of the levels of proteins HuR, APOB-100/48, UQCRB, NDUFB6, and CYCS in mice fed *ad libitum* (AL) and mice fed a calorie-restricted diet (CR). The results showed that CR induced HuR levels mildly. Accordingly, the levels of APOB-100/48, UQCRB, NDUFB6, and CYCS were also modestly induced. These results suggest that HuR may also be involved in regulating the levels of APOB-100/48, UQCRB, NDUFB6, and CYCS in response to CR treatment (Supplementary Figure 15) and have been explained in the 'Discussion' section.

4) *Any metabolomic data available on these tissues?*

Given that hepatocyte-specific HuR deletion influenced the serum levels of HDL and LDL, we are now studying if HuR-regulated lipid metabolism in liver could influence the process of atherosclerosis by using hepatocyte-specific HuR KO mice crossed with APOE^{-/-} mice. The full metabolomic analysis of liver tissue and serum in these mice is underway and would be best presented after in-depth analysis in a dedicated manuscript, to be submitted in the near future. We offer a preview of these data, but hope the reviewer allows us to exclude it from the present manuscript, as we develop it further (Figures 3 and 4, and Tables 1 and 2 for reviewer 1).

Figure 3 for reviewer 1. Liver lipid metabolites analysis in WT and HuR cKO mice.

Serum lipid metabolites in wt vs. HuR cKO (VIP>1)

Figure 4 for reviewer 1. Serum lipid metabolites analysis in WT and HuR cKO mice.

Table 1. Liver metabolites in wt vs. HuR cKO (VIP>1)

Metabolites	VIP value	% Average changes(WT/cKO)
Malic acid	1.6903	90.4
Citric acid	1.6711	61.7
AMP	1.6658	57.6
Succinic acid	1.6106	75.5
Aconitine	1.595	84.4

Table 2. Serum metabolites in wt vs. HuR cKO mice (VIP>1)

Metabolites	VIP value	% Average changes(WT/cKO)
α -ketoglutarate	2.7856	49.0
Aconitine	2.5152	107.0
Succinic acid	1.1712	52.0

Tables 1 and 2 for reviewer 1. Liver and serum metabolites in WT and HuR cKO mice.

5) *What about rescuing with the specific targets to show the importance of these targets?*

Because HuR has been shown to regulate hundreds of target mRNAs, the phenotype seen after abrogating HuR cannot be rescued by ectopically expressing one or several targets,

particularly at a physiologic level. Re-expression of HuR rescued the effect of HuR in HFD-induced NAFLD; we have added results from two additional mice (Figure 5 for reviewer 1), increasing the ‘n’ from 3 to 5 and reflecting this improved analysis in the revised Fig. 8c-e (the means \pm SD and the significance analysis).

Figure 5 for reviewer 1. Two additional HuR cKO mice infected with an adeno-associated virus expressing HuR. **(a)** Western blot analysis to test the effect of HuR re-expression in rescuing the expression of APOB-100, APOB-48, CYCS, NDUFB6, and UQCRB. **(b, c)** The effect of HuR re-expression in liver Chol and TG **(b)** as well as the NDFLD phenotype **(c)** were assessed.

We also tried to rescue the effect of HuR in regulating HFD-induced NAFLD by infecting hepatocytes with an adeno-associated virus expressing the 3'UTR or CR (negative control: a fragment of the coding region of *UQCRB* which does not associate with HuR and did not express the protein of *UQCRB*) fragment of *UQCRB*. As shown in the supplementary Figures 13 and 14, expressing of *UQCRB* 3'UTR, but not *UQCRB* CR, enhanced the effect of HFD in inducing NAFLD but mitigated HFD-induced expression of HuR, APOB-100/48, *UQCRB*, *NDUFB6*, and *CYCS*. Of note, the induction of HuR was also rescued, as HuR can promote the expression of HuR itself (Yi et al., Reduced nuclear export of *HuR* mRNA by HuR is linked to the loss of HuR in replicative senescence. *Nucleic Acids Res.* 2010. 38(5):1547-58).

6) Under these conditions what happens to ROS levels and H2AX?

We appreciate this question. In the revised manuscript, we have included data to show ROS levels, as well as the levels of H2AX in WT and HuR cKO mice in response to HFD (supplementary figure 5). As the reviewer can appreciate, both are elevated in liver of HuR cKO mice.

7) The phenotype is not striking, have the investigators thought about providing another inducer/stressor to the system beyond modulating the diet?

We appreciate the reviewer's comment. A relevant liver inducer/stressor is fibrosis, but the mechanisms for liver fibrosis will be very different from those of NAFLD, as explained by Ge and coworkers in 'Essential Roles of RNA-binding Protein HuR in Activation of Hepatic Stellate Cells Induced by Transforming Growth Factor- β 1'. Sci Rep. 2016).

Therefore, we only tested the relevance of HuR in influencing the levels of targets APOB-100/48, CYCS, NDUFB6, and UQCRB in AL/CR mice. These results have been included in the revised manuscript (Supplementary Figure 15).

8) The adenoviral rescue work for altering the NAFLD phenotype is not convincing as shown, can the investigators show more convincing/dramatic data?

We appreciate this comment and have addressed in the response to point 5.

9) Any more relevant human samples from actual (non-alcoholic fatty liver disease) to show levels of HuR and these targets?

We thank the reviewer for this great suggestion and agree that including these data will increase the clinical significance of our study. We thus sought human samples from the First Affiliated Hospital of Medical School of Zhejiang University, one of the most important hospitals for liver transplantation. Unfortunately, we were denied permission to use these samples for a study of this type, particularly in the large numbers needed to address individual differences. We mention this point in the 'Discussion' section.

Responses to the comments from Reviewer 2:

The goal of this paper was to explore the role of the RNA-binding protein HuR in lipid metabolism and identify potential mechanisms linking HuR and NAFLD. To undertake this goal, the authors created a conditional hepatocyte-specific HuR knockout mouse and evaluated the role of HuR in high-fat diet-induced NAFLD. The major claims of the paper are that HuR associates with key proteins involved in lipid transport, including APOB, UQCRB, and NDUFB6, reduced liver lipid transport, and decreased ATP levels, thereby exacerbating HFD-induced NAFLD.

While HuR has been actively investigated within the context of cancer and aging, its role in lipid metabolism and NAFLD remain unexplored. Therefore, these findings are considered novel and given the results showing a diet-induced effect of HuR, these findings will be of interest to NAFLD researchers. In light of the ubiquitous expression of HuR and its impact on diverse cell activities, the results are also expected to be of interest to the wider community as well.

Injection of HuR cKO mice with adenovirus expressing HuR markedly improved the NAFLD phenotype and elevated levels of APOB-100, ALOB-46, CYCS, and NDUFB6, which lends credence to the major findings. The experimental design is solid and the work convincing, although it would be interesting to see what effect HuR has on actual progression of NAFLD. Results from this study are likely to influence thinking in the field by underscoring a new candidate for diet-induced NAFLD, as well as a new mechanism by which NAFLD develops in the absence of major disruption in liver function or hepatocyte morphology. There are no concerns with the statistical analyses.

One suggestion for this paper is to include a schematic depicting the pathway, with all relevant players, by which HuR contributes to pathogenic features of NAFLD.

We thank the reviewer for his/her strong endorsement of our study. We appreciate the suggestion that we include a schematic depicting the pathway for our studies and have included such a model (Supplementary model 1) in the revised manuscript.

Reviewers' comments second round:

Reviewer #1 (Remarks to the Author):

The authors have done a satisfactory job addressing many of the comments, however still the authors are dismissing the importance of addressing points 4, 7-9 from the original reviewer 1. In an effort to reach a high level study, this reviewer believes these comments should be addressed with experiments and data.

Reviewer #2 (Remarks to the Author):

The authors have responded to the points raised in the previous review of the manuscript.

Point-by-point response to the reviewers

Response to reviewer 1's comments

The authors have done a satisfactory job addressing many of the comments, however still the authors are dismissing the importance of addressing points 4, 7-9 from the original reviewer 1. In an effort to reach a high level study, this reviewer believes these comments should be addressed with experiments and data.

We thank the reviewer for his/her suggestions. His/her concerns are addressed as follows:

4) Any metabolomic data available on these tissues?

We appreciate the Reviewer's request that we give further attention to these points. In the revised manuscript, we have included new metabolomic data (supplementary Figure 6-7). From the extensive liver lipid metabolomic dataset, we focused on the prominent changes in TG (TAG) and CHOL (CE), and displayed them in the supplementary Figure 6-7. Just to be sure, the liver and serum lipid metabolomic data were labeled incorrectly in our point-by-point response letter (WT and cKO were shifted); we have corrected it in the revised Supplementary Figure 5c.

In addition, we performed new liver lipid metabolomic analysis. The new data (shown in Supplementary Figure 5a-b) agree with the earlier results and provide further clarity. As we felt that Reviewer 1 may be also interested in the changes of other species of lipids, the data on lipids other than that of TG and CHOL are included for his/her review:

7) The phenotype is not striking, have the investigators thought about providing another inducer/stressor to the system beyond modulating the diet?

We appreciate this suggestion. In the initial study, we focused on the role of HuR in regulating basal lipid and energy homeostasis (including liver and serum lipid and energy metabolomics; supplementary Figure 6-7). We felt that diet interference was an intervention that was physiologically more meaningful than other inducers/stressors.

In the revised manuscript, we have included data testing the effect metformin administration on the levels of HuR, APOB-100, APOB-40, CYCS, UQCRB, and NDUFB6 (Supplementary Figure 18b). These findings lend further support to the role of HuR in modulating lipid or energy metabolism.

8) The adenoviral rescue work for altering the NAFLD phenotype is not convincing as shown, can the investigators show more convincing/dramatic data?

As we previously responded, HuR is a regulator targeting multiple targets. Therefore, expression one of the targets may be insufficient to rescue the phenotype resulted from HuR knockout. Even so, we also tried to rescue the effect of HuR deletion in aggravating HFD-induced NAFLD by overexpressing CYCS. As shown in Supplementary Figure 17, overexpression of CYCS could rescue the effect of HuR deletion in reducing ATP synthesis (Supplementary Figure 17a-b). However, overexpression of CYCS had no effect in rescuing HuR deletion-aggravated NAFLD phenotype (HFD-induced) (Supplementary Figure 17c-d). These data suggest that overexpression of a single target of HuR may be insufficient to rescue the effect of HuR cKO in aggravating HFD-induced NAFLD.

9) Any more relevant human samples from actual (non-alcoholic fatty liver disease) to show levels of HuR and these targets?

To respond to this question, we studied human samples by both reverse transcription (RT) followed by real-time quantitative (q)PCR and immunohistochemistry analyses. We obtained small amounts of RNA samples (frozen samples) from normal human liver (n=6) and liver from NAFLD patients (n=8). We also obtained slices of normal human liver tissues (n=5) and NAFLD patient liver tissues (n=5). These samples are old samples and kept for longer time. Therefore, they can only be allowed for retrospective study (As we mentioned in previous response letter, it is very difficult to get access to the new samples, since the ethical permit for using human samples are strictly limited in China). Because of the limited RNA amounts, we only analyzed the levels of *Apob* pre-mRNA and mRNA. We were also able to analyze the levels of proteins HuR, APOB, CYCS, UQCRB, and NDUFB6 by immunohistochemistry. The results showed that the *Apob* pre-mRNA levels in NAFLD liver were much lower than those in the normal liver. However, while the levels of *Apob* mRNA in NAFLD patient liver tissues were higher than those in normal human liver tissues, the changes were not significant (likely due to the heterogeneity among the different human samples). However, immunohistochemical analyses revealed that the levels of proteins HuR, CYCS, APOB, UQCRB, and NDUFB6 increased in NAFLD patients. These data support our findings in mice and have been included in the revised Supplementary Information (Supplementary Figure 19).

Reviewer #2 (Remarks to the Author):

The authors have responded to the points raised in the previous review of the manuscript.

We thank the reviewer for his/her positive comments.

Reviewers' comments third round:

Reviewer #1 (Remarks to the Author):

the authors have satisfactorily modified the manuscript

Reviewer #3 (Remarks to the Author):

The metabolomics and lipidomics analysis raises several questions:

The lipids piece refers to SIMCA-P generated VIP scores, but these are not presented in the manuscript. The scales presented in the supplemental are neither described nor are they adequately labelled ... are these standard deviations? How were they calculated? Within class? Within Lipid? Across all groups? Were outliers removed? When were these samples drawn? Were the animals fasted? Are these plasma? Sera? [the method says sera, but also refers to sonication to lyse cells, raising concerns) Were any animals excluded? How were data normalized? The method reports absolute quants, but this is not what is presented. Do animals in their two groups eat at the same time of day?

It is unclear why the lipids were done by two different external labs. It is also unclear why the protocol indicates that cells in sera were broken by ultrasonication – suggesting the wrong protocol was inserted and the document was not proof read sufficiently

Depending on how the analysis was done – which is not clear -- it is unclear whether the metabolomics data really supports the paper (line 164 in main text), or is overfit, as it appears

The metabolomics piece also uses VIP scores, but these datasets are almost certainly too small for those VIP scores to be valid. SIMCA-P reports model r and q values, permutation tests are available, and VIP scores are presented Plus-minus error – none of which is in the current manuscript. How was SIMCA-P run? UV? Winsorized? Transformed?

SIMCA-P data also not shown as described (line 98 in supplemental)

At least in sera, LPE and PE seem affected by HuR – please comment

Comment on intragroup differences, e.g., in TGs

It does not look like much of the data meets the assumptions of a t-test, which is used throughout. The editorial policy suggests individual points shown for $N < 10$ – this does not appear to have been met – and is likely important here. Data distributions do not appear clear.

What was the statistical power for the oxygen consumption experiments? It appears to be underpowered.

Presentation/Minor issues

Confirm the RT window for the Progenesis analysis was ~ 1 sec (that's impressive, but unusual)

The lipids from liver and from sera should be presented in a common format. It's unclear what the dendrogram analysis is teaching us, especially with N's this small

The authors need a table of lipids and their abbreviations

The supplemental section has many, many writing errors, spelling errors, etc, some of which are severe enough to obscure the meaning – this needs to be edited carefully

Reviewer #3 (Remarks to the Author):

The metabolomics and lipidomics analysis raises several questions:

The lipids piece refers to SIMCA-P generated VIP scores, but these are not presented in the manuscript. The scales presented in the supplemental are neither described nor are they adequately labelled ... are these standard deviations? How were they calculated? Within class? Within Lipid? Across all groups? Were outliers removed? When were these samples drawn? Were the animals fasted? Are these plasma? Sera? [the method says sera, but also refers to sonication to lyse cells, raising concerns) Were any animals excluded? How were data normalized? The method reports absolute quants, but this is not what is presented. Do animals in their two groups eat at the same time of day?

We greatly appreciate the reviewer's comments, addressing these concerns will improve greatly our manuscript and correct errors. The purpose of metabolomics analysis in this manuscript is to obtain additional evidence to confirm the changes of lipids shown in the main text (Fig. 3c-d, and fig. 4a). Because we used cKO mice (they need to be similar age, male, fed with HFD at same time, and fresh samples for metabolomics analysis), it is difficult to obtain more than 10 male mice at similar age to do the metabolomics analysis.

Because the animal numbers are not more than 10, we realized that VIP score analysis is not appropriate. We have replaced previous supplementary figures 5-6 with histograms (revised supplementary Figures 5-9), the means \pm SD (standard deviations) from two groups (WT and cKO) as well as the significance were also showed. For the animals, we did not remove the outliers. The mice are not fasted. Male mice (as described in the "methods" section of the main text) of about 8 week old were fed with HFD at the same day for 4 weeks, then the liver tissues or serum (not plasma) were subjected to the analysis (the companies requested fresh tissues or serum for the analysis).

We asked the company for the reason of sonication. We are told that the original method was wrong described. Sonication was used to improve the reaction, not to break the cells. These errors have been changed in the revised "Supplementary Methods".

We did not exclude any animals. The amounts were relative quantitation and normalized by using internal standard (the information has been included in the Supplementary Methods). The animals were fed with HFD at the same day and subjected to the analysis at same time.

It is unclear why the lipids were done by two different external labs. It is also unclear why the protocol indicates that cells in sera were broken by ultrasonication – suggesting the wrong protocol was inserted and the document

was not proof read sufficiently.

Figure 1 for reviewing purpose. Lipid metabolomics analysis from Shanghai Luming Biological Technology Co, LTD exhibited the elevation of TAG in HuR cKO mice. The WT and cKO mice were fed with HFD for 4 weeks.

Originally, we asked Shanghai Luming Biological Technology Co, LTD to do all of the metabolomics analysis. As shown in Figure 1 for reviewing purpose, HuR knockout mice exhibited elevated TAG levels, these data are similar to the data shown in Supplementary Figure 6 (from LipidALL Technologies Co., Ltd). Among all the

metabolomics analysis, the liver lipid metabolomics analysis is the most important one for current study, since lipid accumulation phenotype is very clear. We therefore asked LipidALL Technologies Co., Ltd (Beijing, China) to do one more set of the liver metabolomics analysis [only liver lipids, we used 5:5 because they charged by 10 samples (if it is 12 samples, the cost is same as 20)]. It looks like that the changes of TAG reported from LipidALL Technologies Co., Ltd are more significant and this company also tested the changes of CE (Supplementary Figure 5), which was not tested by the Shanghai Luming Biological Technology Co, LTD. We therefore chose to show the data of liver lipid analysis from LipidALL Technologies Co., Ltd in the Supplementary Information.

The serum lipid compounds were extracted by liquid-liquid extraction with methanol, water and trichloromethane. Ultrasonication was used to improve the reaction. Though not classical, this operation indeed improved a little bit the LC-MS response of metabolites in our detection.

We have contacted the company (Shanghai Luming Biological Technology Co, LTD) and asked the method details. The wrong protocol do insert in the methods. We are so sorry for the mistakes and have revised them in the revised supplementary methods.

Depending on how the analysis was done – which is not clear -- it is unclear whether the metabolomics data really supports the paper (line 164 in main text), or is overfit, as it appears .

As mentioned, we only used 5:5 or 6:6 animals to do the metabolomics and lipidomics analysis. Our major purpose is to obtain further evidence to support the liver and serum data shown in the figures of the main text (Fig. 3c-d, and fig. 4a). However, we agree with the reviewer, the description should be revised to avoid overfit (over interpretation). Therefore, in the revised main text, we described as “By liver and serum lipid and metabonomics analysis, the changes of liver and serum lipid and energy metabolites tended to be consistent with the data shown in Figure 3c-d and Figure 4a (the list of the changed metabolites was shown in Supplementary Figures 5-9)’.

The metabolomics piece also uses VIP scores, but these datasets are almost certainly too small for those VIP scores to be valid. SIMCA-P reports model r and q values, permutation tests are available, and VIP scores are presented Plus-minus error – none of which is in the current manuscript. How was SIMCA-P run? UV? Winsorized? Transformed?

The reviewer is correct, using of the VIP scores are not appropriate because we did not have enough mice (N<10). In the revised manuscript, we reanalyzed the original data and presented the results by using histogram (Supplementary Figure 5-9). Therefore, We do not need data analysis by SIMCA-P or similar software.

SIMCA-P data also not shown as described (line 98 in supplemental)

In the revised manuscript, we presented the data by using histograms. SIMCA-P or similar software was not used.

At least in sera, LPE and PE seem affected by HuR – please comment

In the revised manuscript, the LPE and PE was also included in Supplementary Figure 8.

Comment on intragroup differences, e.g., in TGs

It does not look like much of the data meets the assumptions of a t-test, which is used throughout. The editorial policy suggests individual points shown for N<10 – this does not appear to have been met – and is likely important here. Data distributions do not appear clear.

In the revised manuscript, we reanalyzed the distribution of all of the data needing statistical analysis. Shapiro-Wilk test was used to test the data distribution. A Students' t test was used when Shapiro-Wilk P value > 0.05; A Mann-Whitney U test was used if Shapiro-Wilk P value < 0.05. Only when differences are significant ($p < 0.05$ or 0.01) are indicated. After above changes, Fig. 1d changed from $p < 0.05$ to $p > 0.05$; Supplementary Fig. 18d (TG) changed from $p < 0.01$ to $p < 0.05$). The other significances (p value) are same as previous analysis. The changes have been reflected in the main figures and the the main text. The statistical analysis was also included in the "Methods" section of the main text.

What was the statistical power for the oxygen consumption experiments? It appears to be underpowered.

Because the data in this figure (original Supplementary Figure 11) are not closely related to the major topic of our manuscript, we have removed it from the supplementary information.

Presentation/Minor issues

Confirm the RT window for the Progenesis analysis was ~ 1 sec (that's impressive, but unusual)

We have checked the methods. It is ~ 0.2 min.

The lipids from liver and from sera should be presented in a common format. It's unclear what the dendrogram analysis is teaching us, especially with N's this small.

In the revised manuscript, we presented the lipid and energy data in a common format (histograms).

The authors need a table of lipids and their abbreviations

In the revised manuscript, we have included the abbreviations in the “Supplementary Methods”.

The supplemental section has many, many writing errors, spelling errors, etc, some of which are severe enough to obscure the meaning – this needs to be edited carefully

The supplementary information has been revised and the writing errors have been corrected.

Reviewers' comments fourth round:

Reviewer #3 (Remarks to the Author):

The manuscript is improved by the changes made, and it appears that the TAG data supports the general manuscript, but there remain fundamental concerns. Three examples (other problems almost certainly exist, but these are severe enough as to render the manuscript difficult to review:

In the response to the review, it is noted that...(page 3)... It looks like that the changes of TAG reported from LipidALL Technologies Co., Ltd are more significant. In other words, it sounds like the authors selectively chose data to fit their desired hypothesis. It is concerning that the two companies did not agree, and it is unclear why the authors did not attempt to clarify this problem. It also suggests that there are major quality control concerns.

The rationale for using a T-test...from the authors rebuttal letter... Shapiro-Wilk test was used to test the data distribution. A Students' t test was used when Shapiro-Wilk P value > 0.05; A Mann-Whitney U test was used if Shapiro-Wilk P value < 0.05. Based on what I can find in the literature, the Shapiro Wilk test has a power of <0.1 with the N's of 5 and 6 used here, and an N of >100 is needed to reach a power of 0.8. Thus, this test is not valid, and, again, T-tests are not appropriate, and speaks to a fundamental error in the statistical analysis of the data

If one looks at liver citric acid (supplemental Figure 9, Panel A) The value given for SD is 1.0+ 0.4 or so for hollow bars and 1.6 +/- ~0.4 for the black bars.. A quick excel analysis of the data provided suggests the real values are 1 +/- 0.9 and 1.6+/-0.8, assuming that the wildtype is the hollow bar. This single spot check suggests there are fundamental errors in the basic analysis done.

Also, note that the plots in supplemental Figure 5 and 6 are typically called column or bar plots, they are not histograms. This error, while itself relatively small, speaks to the need to have the details of all -omics analysis reviewed by people with appropriate expertise

Point-by-point response to Reviewer #3 (Remarks to the Author):

The manuscript is improved by the changes made, and it appears that the TAG data supports the general manuscript, but there remain fundamental concerns. Three examples (other problems almost certainly exist, but these are severe enough as to render the manuscript difficult to review:

We very much appreciate the additional time that this reviewer has dedicated to ensuring our manuscript is fully ready. His/her helpful comments have been addressed below.

In the response to the review, it is noted that...(page 3)... It looks like that the changes of TAG reported from LipidALL Technologies Co., Ltd are more significant. In other words, it sounds like the authors selectively chose data to fit their desired hypothesis. It is concerning that the two companies did not agree, and it is unclear why the authors did not attempt to clarify this problem. It also suggests that there are major quality control concerns.

We appreciate the reviewer's concern. In our last point-by-point response letter, we stated "*The purpose of the metabolomic analysis in this manuscript is to obtain additional evidence to confirm the changes of lipids shown in the main text (Fig. 3c-d, and fig. 4a) and the data of TAG from both companies are very similar*". We included the TAG data from the other company for consideration by the reviewer. The data from both companies supported the observations that HuR knockout led to accumulation of lipids in the liver. It was not possible to merge the data from the two companies for significant analysis, since they were collected from different groups of animals and at different times. Even if samples from different groups of mice collected at different times had been analyzed by same company, the data would also have been different. What is important, in our humble opinion, is that the trend from both companies is same. In this case, the trend of TAG changes from both companies is consistent.

The rationale for using a T-test...from the authors rebuttal letter... Shapiro-Wilk test was used to test the data distribution. A Students't test was used when Shapiro-Wilk P value > 0.05; A Mann-Whitney U test was used if Shapiro-Wilk P value < 0.05. Based on what I can find in the literature, the Shapiro Wilk test has a power of <0.1 with the N's of 5 and 6 used here, and an N of >100 is needed to reach a power of 0.8. Thus, this test is not valid, and, again, T-tests are not appropriate, and speaks to a fundamental error in the statistical analysis of the data

The reviewer is correct. Because the numbers of mice was small (n=5-6 or 5-10), the distribution cannot be easily determined, and a nonparametric test should be appropriate. Therefore, in the revised manuscript, a Mann-Whitney U test was used to assess significance from the data obtained from mouse experiments.

For data from cell experiments, with 3 independent biological repeats from the same cell lines and treatments, it is not necessary to analyze the distribution of the data. In this case, Students' t test is widely used to assess significance [see for example Loregger et al. Nature Communications 11, 1128 (2020); Corbert et al. Nature Communications 11, 454 (2020); Fu et al. Nature Communications 11, 438 (2020); Zhao et al. Nature Communications 11, 341 (2020); Li et al. Nature Communications 10, 2375 (2019)]. Therefore, to assess significance of data using cells, we have retained Student's t test in the revised manuscript.

The changes of the significance after above mentioned analysis are reflected in the revised submission.

If one looks at liver citric acid (supplemental Figure 9, Panel A) The value given for SD is 1.0+ 0.4 or so for hollow bars and 1.6 +/- ~0.4 for the black bars.. A quick excel analysis of the data provided suggests the real values are 1 +/- 0.9 and 1.6+/-0.8, assuming that the wildtype is the hollow bar. This single spot check suggests there are fundamental errors in the basic analysis done.

We appreciate the reviewer's catching this mistake. Indeed, some of the error bars shown in the last version were SEM; the software shifted SD to SEM by default and we did not notice it. We have corrected this problem in the revised version. We have checked every graph carefully and have ensured that we use SD throughout.

Also, note that the plots in supplemental Figure 5 and 6 are typically called column or bar plots, they are not histograms. This error, while itself relatively small, speaks to the need to have the details of all -omics analysis reviewed by people with appropriate expertise.

We thank the reviewer for pointing out this wrong description. In the revised manuscript, these Figures are presented as bar plots, not histograms.